# Representation Preserving Multiclass Agnostic to Realizable Reduction

Steve Hanneke [1]   Qinglin Meng [1]   Amirreza Shaeiri [1]

## Abstract

We study the problem of multiclass classification when the number of labels can be unbounded within the PAC learning framework (Valiant, 1984). Our main contribution is a theory that demonstrates a *simple* and *elegant* agnostic to realizable reduction for this framework. This resolves an open problem raised by the recent work of (Hopkins et al., 2022). Notably, our result is the first *representation preserving* multiclass agnostic to realizable reduction, in contrast with the compression based approach of the work of (David et al., 2016). Furthermore, our main theorem is stated in an abstract framework, called "Unified PAC Learning", which encompasses a range of frameworks, including multiclass PAC learning, list PAC learning, and multilabel PAC learning. In addition, we explore representation preserving reductions to the realizable setting for two noise models, namely Massart noise and Tsybakov noise, in the multiclass PAC learning framework. We believe our technique may find other applications in ensuing studies of theoretical machine learning.

## 1. Introduction

In many frameworks within the field of statistical learning theory, a surprising equivalence emerges between *realizable* and *agnostic* learnability, where both are characterized by the same quantity despite their inherent difference. This phenomenon encompasses fundamental frameworks, including binary PAC learning, multiclass PAC learning, list PAC learning, binary online learning, and multiclass online learning, among others. Nonetheless, the proof techniques employed to establish the realizable and agnostic results can differ significantly in many of these frameworks. This observation raises an intriguing question: Do we possess a *unified* theory that underlies these equivalences? Even more optimistically, could there exist a *simple* and *elegant* unified theory, both in terms of the analysis and reduction properties, that accounts for these equivalences?

The seminal work of (David et al., 2016) provides a basis for developing a unified theoretical framework. In fact, for almost a decade, their sample compression scheme based approach, which involves boosting, serves as the only tool for theorists to prove agnostic learnability in frameworks such as partial PAC learning, multiclass PAC learning, and list PAC learning, where the number of labels can be unbounded in the latter two cases. However, this approach can produce more complex functions than those outputted by a realizable learner due to its reliance on the boosting technique, which combines multiple realizable learners. In particular, consider the space of possible outputting functions in the realizable setting. Then, in the agnostic setting, the resulting functions based on this approach do not necessarily belong to that space.

In addition to the mentioned reduction, the recent work of (Hopkins et al., 2022) provides an affirmative answer to the aforementioned questions across various frameworks. More specifically, they introduced a three-line algorithm accompanied by a relatively simple proof technique, which effectively demonstrates the existence of reductions in many frameworks. Moreover, their algorithm uncovers some previously unknown equivalences between realizable and agnostic learnability. However, despite these advancements, their approach does not extend to important problems, such as multiclass PAC leaning with possibly unbounded label space framework. [1] We suggest readers refer to section 2 for a detailed discussion on the significance of this setting. Consequently, they posed the following open question: Is there a unifying yet still simple and elegant technique that can also address multiclass PAC learning with possibly infinite label space framework?

In this work, we provide POSITIVE ANSWER to their open question. Specifically, we demonstrate that a clever modification of their three-line algorithm indeed effectively reduces agnostic learnability to realizable learnability in frameworks such as multiclass learning with potentially unbounded label spaces.

---

[1]Department of Computer Science, Purdue University, West Lafayette, IN, USA. Correspondence to: Steve Hanneke <steve.hanneke@gmail.com>.

*Proceedings of the $42^{nd}$ International Conference on Machine Learning*, Vancouver, Canada. PMLR 267, 2025. Copyright 2025 by the author(s).

---

[1]This is discussed formally in Appendix A.

Additionally, our algorithm possesses a UNIQUE PROPERTY inherited from the algorithm of (Hopkins et al., 2022). To illustrate, consider the space of possible outputting functions in the realizable setting. Our reduction always outputs a function that belongs to that space, which is in sharp contrast to the approach of (David et al., 2016) as we discussed. Going further, our reduction even preserves data-dependent properties of the output function of the realizable setting, such as the geometric margin used in Support Vector Machines, rather than merely producing a function from the mentioned space. As a result, our work relates to the line of research focused on learning with simple predictors that has gained significant attention in recent years. For instance, the work of (Hanneke et al., 2021) explored this topic in the context of online learning, while (Aden-Ali et al., 2024; Bousquet et al., 2020) studied this topic in the context of binary PAC learning. Therefore, our technique provides a valuable new tool for theorists to utilize in their problems when required.

In fact, we state our main theorem within a new abstract framework, which we call UNIFIED PAC LEARNING. This framework encompasses a range of frameworks, multiclass PAC learning (for instance, see the work of (Brukhim et al., 2022)), list PAC learning (for instance, see (Charikar & Pabbaraju, 2023)), and multilabel PAC learning. As a result, our finding reveals some new previously unknown equivalences and offers new insights into the relationship between realizable and agnostic learnability. We believe that Unified PAC Learning offers a promising avenue for further exploration of realizable learnability in future research.

Subsequently, we delve into the exploration of NOISE MODELS, which occupy an intermediate position between the realizable and agnostic settings; see (Tsybakov, 2004; Massart & Nédélec, 2006; Agarwal, 2013; Hanneke & Yang, 2015) for more details. In particular, we focus on two very popular noise models, namely Massart noise and Tsybakov noise, for the multiclass PAC learning framework. Using almost the same technique, we demonstrate that focusing on noise models, rather than the agnostic setting, allows for an improvement in the sample complexity.

Finally, we extend our main theorem on the Unified PAC Learning framework to address PARTIAL CONCEPT CLASSES as well. We recommend readers refer to the work of (Alon et al., 2022) for a comprehensive discussion on the importance of this setting. While the same algorithm can be applied to partial concept classes, the analysis becomes more involved. Notably, the work of (Hopkins et al., 2022) also addresses this setting; however, they adopt a different definition of agnostic learnability. In contrast, we employ the definition provided in the recent work of (Alon et al., 2022) on partial concept classes, which is more favorable in this context.

## 1.1. Overview of the Main Result and Technique

In the following subsection, we provide an overview of our main result along with a summary of the primary proof technique in our work. Before proceeding, let us informally define the Unified PAC Learning framework.

In the Unified PAC Learning framework, we have an instance space $\mathcal{X}$, an output space $\mathcal{Y}$, an action space $\mathcal{A}$, and a loss function $\mathcal{L} : \mathcal{A} \times \mathcal{Y} \to \{0, 1\}$. A predictor is a function from $\mathcal{X}$ to $\mathcal{A}$. Moreover, we have a set of predictors $\mathcal{C}$ as a concept class. In addition, we have another set of predictors $\mathcal{H}$ as a hypothesis class. As a result, a 6-tuple consisting of $\mathcal{X}, \mathcal{Y}, \mathcal{A}, \mathcal{L}, \mathcal{C}$, and $\mathcal{H}$ specifies an instance of this framework. Importantly, both the output space $\mathcal{Y}$ and the action space $\mathcal{A}$ can be unbounded.

Furthermore, in this framework, a learning algorithm maps a sequence of instance-output pairs of any size to a predictor. Also, a data distribution is defined on $\mathcal{X} \times \mathcal{Y}$. Based on these, we adopt the definitions of loss for a given data distribution and predictor, realizable data distribution, realizable PAC learnability, and agnostic PAC learnability within this framework. Importantly, in both the definitions of realizable PAC learnability and agnostic PAC learnability, we require that the image of the learning algorithm be a subset of $\mathcal{H}$. See section 3 for formal notations and definitions.

In this paragraph, we provide a concise explanation of how our framework can be utilized to encompass various settings, illustrated through two examples. For instance, by defining $\mathcal{Y}$ as a set of labels and $\mathcal{A}$ as $\{a \mid a \subseteq \mathcal{Y}, |a| \leq \mathbf{L}\}$ for some $\mathbf{L} \in \mathbb{N}, \mathbf{L} \leq |\mathcal{Y}|$, we can effectively capture the list PAC learning framework of (Charikar & Pabbaraju, 2023). Indeed, $\mathcal{A}$ can also be interpreted as a set of labels, then by specifying $\mathcal{Y}$ as $\{y \mid y \subseteq \mathcal{A}, |y| \leq \mathbf{L}\}$ for some $\mathbf{L} \in \mathbb{N}, \mathbf{L} \leq |\mathcal{A}|$, it becomes possible to accommodate the multilabel PAC learning framework. It is worth noting that this definition is a special case of the general multilabel setting in practice, which we find particularly interesting as it serves as a dual formulation of list learning. For further details, refer to section 4.

Now, we informally present our main theorem in the current work.

**Theorem 1.1** (Realizable Unified PAC Learnability $\rightarrow$ Agnostic Unified PAC Learnability)**.** *Let $\mathcal{Q} = (\mathcal{X}, \mathcal{Y}, \mathcal{A}, \mathcal{L}, \mathcal{C}, \mathcal{H})$ be an instance of the Unified PAC Learning framework. If $\mathcal{Q}$ is realizable PAC learnable, then $\mathcal{Q}$ is agnostic PAC learnable.*

The key idea behind the proof of the above theorem is to run the realizable learning algorithm on each subset of samples, followed by applying empirical risk minimization (ERM) on the resulting finite collection of predictors (1). Moreover, by applying a concentration bound to this finite set

of predictors, combined with the error rate guarantee from the realizable PAC learnability for the predictors generated from each subset, we can get the guarantee of agnostic learnability. See section 4 for more details.

As a direct corollary of the above theorem, one can derive a reduction from agnostic PAC learnability to realizable PAC learnability for the multiclass PAC learning framework, as described in subsection 4.2. Additionally, our result reveals new equivalences, including previously unknown equivalence between the realizable multilabel PAC learnability and the agnostic multilabel PAC learnability, as discussed in subsection 4.4.

Furthermore, building upon the work of (David et al., 2016), we establish another reduction for the Unified PAC Learning framework. While this reduction is more favorable from the sample complexity perspective, crucially, it does not retain properties such as representation preserving. In particular, given an accuracy parameter $\epsilon \in (0, 1)$, our agnostic result involves a factor of $1/\epsilon^3$, compared to the more desirable factor of $1/\epsilon^2$ that can be obtained using a sample compression scheme based proof. For further details, refer to section 4 and Appendix C. However, for every $n, m \in \mathbb{N}$, we give an example of an instance of the Unified PAC Learning framework $\mathcal{Q} = (\mathcal{X}, \mathcal{Y}, \mathcal{A}, \mathcal{L}, \mathcal{C}, \mathcal{H})$ such that $|\mathcal{X}| = n$, $|\mathcal{Y}| = |\mathcal{A}| = |\mathcal{C}| = |\mathcal{H}| = m$ where we have: for every $\epsilon, \delta \in (0, 1)$, there exists a $\Omega(1/\epsilon^2)$ gap between the optimal sample complexity of the realizable PAC learning with parameters $\epsilon$ and $\delta$ and the optimal sample complexity of the agnostic PAC learning with the same parameters. In particular, this example is within multiclass PAC learning framework. Moreover, it builds upon the concept class of the countably infinite collection of constant functions over some domain. For further details, refer to section 6.

### 1.2. Organization

The remainder of this paper is structured as follows. In section 2, we discuss a broader range of related works. Then, in section 3, we present the formal notations and definitions. Subsequently, in section 4, we establish our main reduction along with some corollaries. Following this, in section 5, we explore noise models in the multiclass PAC learning framework. Finally, in section 7, we conclude the manuscript.

## 2. Related Work

**PAC Learning.** The Probably Approximately Correct (PAC) learning framework, introduced by (Valiant, 1984), has been a cornerstone in the field of statistical learning theory. (Blumer et al., 1989; Vapnik & Chervonenkis, 2015; Valiant, 1984; Vapnik, 2006) characterizes learnable classes within the binary PAC learning framework in the realizable

setting via a combinatorial parameter called the VC dimension. This result was later extended to the agnostic setting by (Haussler, 1992). Since then, PAC learning has been extensively studied in various learning theoretic settings.

**Multiclass Classification.** A large body of theoretical research has been devoted to studying multiclass classification in different frameworks. This includes contributions from (Natarajan & Tadepalli, 1988; Natarajan, 1989; Ben-David et al., 1992; Haussler & Long, 1995; Rubinstein et al., 2006; Daniely et al., 2011; 2012; Daniely & Shalev-Shwartz, 2014; Brukhim et al., 2021). Nevertheless, the combinatorial characterization of multiclass classification, when the number of labels can be unbounded, within Valiant's PAC learning framework, has remained an open question until recently, even in the realizable setting. The seminal work of (Brukhim et al., 2022) addressed this gap. Furthermore, the same dimension also characterizes the agnostic version of this problem (David et al., 2016). Also, see the recent work of (Hanneke et al., 2023).

There are several reasons motivating this interest in multiclass classification when the number of labels can be unbounded. First, in multiclass settings, guarantees should ideally not depend on the number of labels, even when finite. Second, mathematical concepts that involve infinity often offer clearer and more elegant insights. Finally, from a practical standpoint, many critical machine learning tasks require classification into very large label spaces. This includes the image object recognition task.

## 3. Preliminaries

In this section, we introduce the necessary notations and definitions that will be used throughout the remainder of this work.

### 3.1. Notations

We begin by introducing some necessary notation before describing and analyzing our algorithm. Fix a non-empty set $\mathcal{X}$ equipped with a $\sigma$-algebra specifying the measurable subsets as an instance space. Fix a non-empty set $\mathcal{Y}$ as an output space. Also, fix a non-empty set $\mathcal{A}$ as an action space. In addition, fix a loss function $\mathcal{L} : \mathcal{A} \times \mathcal{Y} \rightarrow \{0, 1\}$. A predictor is a function from $\mathcal{X}$ to $\mathcal{A}$. With this in mind, fix a non-empty set of predictors $\mathcal{C}$ as a concept class. Additionally, fix another set of predictors $\mathcal{H}$ as a hypothesis class. Notably, we implicitly assume standard measurability assumptions on $\mathcal{C}$ and $\mathcal{H}$. In particular, a 6-tuple $\mathcal{Q} = (\mathcal{X}, \mathcal{Y}, \mathcal{A}, \mathcal{L}, \mathcal{C}, \mathcal{H})$ presents an instance of the Unified PAC Learning framework. A learning algorithm $\mathbf{A}$ is a mapping from $(\mathcal{X} \times \mathcal{Y})^*$ to a predictor $\hat{h}$. Also, a data distribution $\mathcal{D}$ is a probability measure on $(\mathcal{X} \times \mathcal{Y})$. Finally, we use $\mathcal{O}(\cdot)$, $o(\cdot)$, $\Omega(\cdot)$, $\omega(\cdot)$, and $\Theta(\cdot)$ as standard notations of them in the theoretical computer science. We also

use $\widetilde{\mathcal{O}}(\cdot)$, $\widetilde{\Omega}(\cdot)$, $\widetilde{\Theta}(\cdot)$ to exclude logarithmic factors as well as constant coefficients.

## 3.2. Definitions

Given the above notations, we make the definitions for true risk and empirical risk. For a data distribution $\mathcal{D}$ and a predictor $\hat{h}$, define the true risk of $\hat{h}$ with respect to the data distribution $\mathcal{D}$ as follows,

$$L_{\mathcal{D}}(\hat{h}) := \mathbb{P}_{(x,y)\sim\mathcal{D}}[\mathcal{L}(\hat{h}(x), y)].$$

Based on this, we say that a data distribution $\mathcal{D}$ is realizable by a concept class $\mathcal{C}$ if there exists $c^* \in \mathcal{C}$ such that $L_{\mathcal{D}}(c^*) = 0$. Similarly, say that a data distribution $\mathcal{D}$ is realizable by a hypothesis class $\mathcal{H}$, if $\inf_{h\in\mathcal{H}} L_{\mathcal{D}}(h) = 0$ holds. For simplicity, we will suppose the infimum $\inf_{c\in\mathcal{C}} L_{\mathcal{D}}(c)$ is actually achieved by a concept $c^* \in \mathcal{C}$. [2]

Next, a data set of size $m$ for some $m \in \mathbb{N}$ is a subset of $(\mathcal{X} \times \mathcal{Y})$ of size $m$. For such a data $\mathcal{S} = \{(x_1, y_1), (x_2, y_2), \ldots, (x_m, y_m)\}$ and a predictor $\hat{h}$, define the empirical risk of $\hat{h}$ with respect to the data distribution $\mathcal{S}$ as follows,

$$L_{\mathcal{S}}(\hat{h}) := \frac{1}{m} \sum_{i=1}^{m} \mathcal{L}(\hat{h}(x_i), y_i). \tag{1}$$

We say a data set $S$ is realizable by a predictor $\hat{h}$ if $L_S(\hat{h}) = 0$. Subsequently, we present the definitions of realizable and agnostic PAC learnability for this framework.

**Definition 3.1** (Unified PAC Learning). We say that $\mathcal{Q} = (\mathcal{X}, \mathcal{Y}, \mathcal{A}, \mathcal{L}, \mathcal{C}, \mathcal{H})$ is realizable Unified PAC learnable, if for every $\epsilon, \delta \in (0, 1)$, there exists a finite $\mathcal{M}^{\mathrm{RE}}(\epsilon, \delta) \in \mathbb{N}$ and a learning algorithm $\mathbf{A}$ with $\mathrm{Im}(\mathbf{A}) \subseteq \mathcal{H}$ such that, for every distribution $\mathcal{D}$ on $(\mathcal{X} \times \mathcal{Y})$ realizable w.r.t. $\mathcal{C}$, for $S \sim \mathcal{D}^{\mathcal{M}^{\mathrm{RE}}(\epsilon,\delta)}$, with probability at least $1 - \delta$,

$$L_{\mathcal{D}}(\mathbf{A}(S)) \leq \epsilon.$$

The value $\mathcal{M}^{\mathrm{RE}}(\epsilon, \delta)$ is called the sample complexity of $\mathbf{A}$, and the optimal sample complexity[3] is the minimum achievable value of $\mathcal{M}^{\mathrm{RE}}(\epsilon, \delta)$ for every given $\epsilon, \delta$.

**Definition 3.2** (Unified Agnostic PAC Learning). We say that $\mathcal{Q} = (\mathcal{X}, \mathcal{Y}, \mathcal{A}, \mathcal{L}, \mathcal{C}, \mathcal{H})$ is agnostic unified PAC learnable, if for every $\epsilon, \delta \in (0, 1)$, there exists a finite $\mathcal{M}^{\mathrm{AG}}(\epsilon, \delta) \in \mathbb{N}$ and a learning algorithm $\mathbf{A}$ with $\mathrm{Im}(\mathbf{A}) \subseteq \mathcal{H}$ such that, for every distribution $\mathcal{D}$ on $(\mathcal{X} \times \mathcal{Y})$, for $S \sim \mathcal{D}^{\mathcal{M}^{\mathrm{AG}}(\epsilon,\delta)}$, with probability at least $1 - \delta$,

$$L_{\mathcal{D}}(\mathbf{A}(S)) \leq L_{\mathcal{D}}(c^*) + \epsilon,$$

---

[2]If not, for any fixed $\epsilon > 0$, we can choose $c^*$ such that $L_{\mathcal{D}}(c^*) \leq \inf_{c\in\mathbb{C}} L_{\mathcal{D}}(c) + \epsilon/2$.

[3]For brevity, we refer to the optimal sample complexity as sample complexity.

---

**Algorithm 1** Agnostic to Realizable Reduction

**Input:** Concept Class $\mathcal{C}$, Hypothesis class $\mathcal{H}$, Realizable PAC-Learner $\mathbf{A}$, Labeled Sample $S$.
1: Divide $S$ into two parts $V$ and $T$.
2: Run $\mathbf{A}$ over all subsets of $V$ to get:

$$\mathcal{H}_V := \{\mathbf{A}(S') \mid S' \subseteq V\}$$

3: Return the hypothesis in $\mathcal{H}_V$ with lowest empirical error over $T$.

---

where $c^*$ is the optimal concept in the concept class $\mathcal{C}$. The value $\mathcal{M}^{\mathrm{AG}}(\epsilon, \delta)$ is called the sample complexity of $\mathbf{A}$, and the optimal sample complexity is the minimum achievable value of $\mathcal{M}^{\mathrm{AG}}(\epsilon, \delta)$ for every given $\epsilon, \delta$.

## 4. Reduction

We present the unified agnostic-to-realizable reduction in subsection 4.1. The applications of this reduction will be discussed in subsections 4.2, 4.3, and 4.4.

### 4.1. General Reduction

The following theorem demonstrates a representation preserving reduction from agnostic to realizable setting within the unified PAC learning framework.

**Theorem 4.1** (Agnostic $\rightarrow$ Realizable ). *Let* $\mathbf{A}$ *be an unified realizable learner for a learning instance* $\mathcal{Q} = (\mathcal{X}, \mathcal{Y}, \mathcal{A}, \mathcal{L}, \mathcal{C}, \mathcal{H})$ *with sample complexity* $\mathcal{M}^{\mathrm{RE}}(\epsilon, \delta)$. *Then Algorithm 1 is also an unified agnostic learner for* $\mathcal{Q} = (\mathcal{X}, \mathcal{Y}, \mathcal{A}, \mathcal{L}, \mathcal{C}, \mathcal{H})$ *with sample complexity:* $\mathcal{M}^{\mathrm{AG}}(\epsilon, \delta) =$

$$\mathcal{O}\left(\frac{1}{\epsilon^2}\left(\max_{\mu\in[\epsilon/2,1]} \frac{\mathcal{M}^{\mathrm{RE}}(\epsilon/(2\mu), \delta/3)}{\mu} + \log\left(\frac{1}{\delta}\right)\right)\right).$$

We emphasis that Theorem 4.1 presents the first *representation preserving* reduction from agnostic to realizable within unified PAC learning regime. Its key application to multiclass learning further highlights the first such reduction in this domain.

With this settled, we now proceed to prove Theorem 4.1. The analysis naturally divides into two parts, corresponding to steps 2 and 3 of Algorithm 1, respectively. In the first part, we will show that $\mathcal{H}_V$, the set of output hypothesis corresponding to running the realizable learner $\mathbf{A}$ over all subsets $S'$ of $V$, contains a hypothesis $h'$ which is close to the optimal concept $c^*$ in $\mathcal{C}$. More formally, we have the following lemma.

**Lemma 4.2.** *For any distribution* $\mathcal{D}$ *over* $(\mathcal{X} \times \mathcal{Y})$*, with probability at least* $1 - 2\delta/3$*, there exists* $h' \in \mathcal{H}_V$ *which satisfies:*

$$L_{\mathcal{D}}(h') \leq L_{\mathcal{D}}(c^*) + \epsilon/2,$$

*where $c^*$ is the optimal concept in $\mathcal{C}$.*

Once we prove this lemma, the second part is to show that Step 3, an empirical risk minimization process on $\mathcal{H}_V$ over labeled sample $T$, gives the desired agnostic hypothesis. Since $\mathcal{H}_V$ is finite, a standard Chernoff bound gives that with probability at least $1 - \delta/3$, the empirical risk $L_T(h)$ of every hypothesis $h$ in $\mathcal{H}_V$ is $\epsilon/4$-close to its true risk $L_{\mathcal{D}}(h)$, as long as $T$ is sufficiently large. Combining these two parts, Algorithm 1 returns a hypothesis with at most $L_{\mathcal{D}}(c^*) + \epsilon$ error rate with high probability.

It remains to prove Lemma 4.2. The key observation is that the output $\mathbf{A}(S^*)$ of running $\mathbf{A}$ over the subset $S^*$ realizable by $c^*$ is close to $c^*$.

*Proof of Lemma 4.2.* Since the optimal concept $c^*$ may not be realizable with respect to distribution $\mathcal{D}$ over $(\mathcal{X} \times \mathcal{Y})$. We divide the sample space $(\mathcal{X} \times \mathcal{Y})$ into two parts according to whether realizable with respect to $c^*$ and $\mathcal{D}$. Let $(\mathcal{X} \times \mathcal{Y})^*$ denote the realizable part and let $\mathcal{D}^*$ denote the restriction of $\mathcal{D}$ over it. Similarly, let $\bar{\mathcal{D}}^*$ denote the restriction of $\mathcal{D}$ over the complement, that is $(\mathcal{X} \times \mathcal{Y}) \backslash (\mathcal{X} \times \mathcal{Y})^*$. With this in mind, let $\mu^*$ denote the probability mass of $\mathcal{D}$ on $(\mathcal{X} \times \mathcal{Y})^*$. Since we are restricting our attention to classification error, we can decompose the true risk of $c^*$ over $\mathcal{D}$ as:

$$L_{\mathcal{D}}(c^*) = \mu^* L_{\mathcal{D}^*}(c^*) + (1 - \mu^*) L_{\bar{\mathcal{D}}^*}(c^*) = 1 - \mu^*,$$

where the second step is by the decomposition of the sample space $(\mathcal{X} \times \mathcal{Y})$, and the last step follows from the observation that $L_{D^*}(c^*) = 0$ and $L_{\bar{\mathcal{D}}^*}(c^*) = 1$ always hold by definition. To get a hypothesis within $\epsilon/2$ true risk of $L_{\mathcal{D}}(c^*)$, we claim that it is sufficient to prove $\mathcal{H}_V$ contains some $h$ with true risk $\epsilon/2\mu^*$ over $\mathcal{D}^*$, that is some $h$ satisfying:

$$L_{\mathcal{D}^*}(h) \leq \epsilon/(2\mu^*). \tag{2}$$

This follows from a similar analysis of $L_{\mathcal{D}}(h^*)$ above. We can decompose $L_{\mathcal{D}}(h)$ as:

$$L_{\mathcal{D}}(h) = \mu^* L_{\mathcal{D}^*}(h) + (1 - \mu^*) L_{\bar{\mathcal{D}}^*}(h) \leq \epsilon/2 + L_{\mathcal{D}}(c^*).$$

It remains to prove the claim that $\mathcal{H}_V$ contains a hypothesis satisfying Equation(2) with high probability. Notice that $S^*$ can be seen as drawn from the distribution $\mathcal{D}^*$. Then, by the definition of the realizable learning, on the labeled sample $S^* \sim D^*$ of size $\mathcal{M}^{\mathrm{RE}}(\epsilon/(2\mu^*), \delta/3)$, the realizable learner $\mathbf{A}$ will output $h_{S^*}$ satisfying:

$$L_{\mathcal{D}^*}(h_{S^*}) \leq \epsilon/(2\mu^*).$$

with probability at least $1 - \delta/3$.

Then we just need to draw a large enough sample $V$ to make sure that the size of $S^*$ is at least $\mathcal{M}^{\mathrm{RE}}(\epsilon/(2\mu^*), \delta/3)$. By a Chernoff bound, it is enough

to draw $\mathfrak{c}\mathcal{M}^{\mathrm{RE}}(\epsilon/(2\mu^*), \delta/3)/\mu^*$ data points to achieve this for some constant $\mathfrak{c} > 0$. Since we do not know $\mu^*$, we need to draw $\mathfrak{c} \max_{\mu \in [\epsilon/2, 1]} \{\mathcal{M}^{\mathrm{RE}}(\epsilon/(2\mu), \delta/3)/\mu\}$ data points to ensure this claim holds (if $\mu^* < \epsilon/2$, note that any hypothesis will give a valid solution). By a union bound, we have that this overall process holds with probability at least $1 - 2\delta/3$. $\square$

*Proof of Theorem 4.1.* By Lemma 4.2, with probability at least $1 - 2\delta/3$, $\mathcal{H}_V$ contains a hypothesis $h_{S^*}$ such that :

$$L_{\mathcal{D}}(h_{S^*}) \leq L_{\mathcal{D}}(c^*) + \epsilon/2.$$

We can now use standard empirical risk minimization bounds on $\mathcal{H}_V$ to find a hypothesis with true risk at most $L_{\mathcal{D}}(c^*) + \epsilon$. A Chernoff and union bounds imply that with probability at least $1 - \delta/3$, the empirical risk of every hypothesis in $\mathcal{H}_V$ is at most $\epsilon/4$ away from its true risk on a sample of size $\mathcal{O}(\log(|\mathcal{H}_V|/\delta)/\epsilon^2)$. Since $h_{S^*}$ has true risk $L_{\mathcal{D}}(h_{S^*})$ at most $L_{\mathcal{D}}(c^*) + \epsilon/2$, its empirical risk $L_T(h_{S^*})$ is at most $L_{\mathcal{D}}(c^*) + 3\epsilon/4$. Then we can be sure there exists a hypothesis in $\mathcal{H}_V$ with empirical risk at most $L_{\mathcal{D}}(c^*) + 3\epsilon/4$, and by the above guarantee any hypothesis in $\mathcal{H}_V$ with empirical risk at most $L_{\mathcal{D}}(c^*) + 3\epsilon/4$ has true risk at most $L_{\mathcal{D}}(c^*) + \epsilon$.

Combining Lemma 4.2 and the above analysis of empirical risk minimization, we have that with probability at least $1 - \delta$ over the entire process, the output of Algorithm 1 is a desired agnostic learner. The sample complexity follows from noting that $|\mathcal{H}_V|$ is at most $2^{|V|}$ with $|V| = \mathcal{O}(\max_{\mu \in [\epsilon/2, 1]} \{\mathcal{M}^{\mathrm{RE}}(\epsilon/(2\mu), \delta/3)/\mu\})$ due to Lemma 4.2. $\square$

### 4.2. Multiclass Learning

Building on Theorem 4.1, we can readily adapt it to multiclass learning setting with the action space $\mathcal{A}$ equal to the infinite label space $\mathcal{Y}$, and $\mathcal{L}$ represents the classification error. In this way, a unified PAC learning instance $\mathcal{Q} = (\mathcal{X}, \mathcal{Y}, \mathcal{A}, \mathcal{L}, \mathcal{C}, \mathcal{H})$ can be specified as a multiclass learning instance $(\mathcal{X}, \mathcal{Y}, \mathcal{C}, \mathcal{H})$. Then Theorem 4.1 can be applied to this learning setting, together with Theorem 1 and Algorithm 1 in Brukhim et al. (2022) we have the following corollary.

**Corollary 4.3** (Agnostic $\rightarrow$ Realizable (Multiclass Learning))**.** *If $\mathcal{H}$ has finite DS dimension $d$, Algorithm 1 is an agnostic multiclass learner with sample complexity:*

$$\mathcal{M}^{\mathrm{AG}}(\epsilon, \delta) = \tilde{\mathcal{O}}\left(\frac{d^{3/2} + \log(1/\delta)}{\epsilon^3}\right).$$

### 4.3. List Learning

We can also adapt Theorem 4.1 to $\mathbf{L}$-list learning setting with the action space $\mathcal{A} = \{Y \subseteq \mathcal{Y}, |Y| \leq \mathbf{L}\}$. The concept class $\mathcal{C}$ consists of functions $c : \mathcal{X} \rightarrow \mathcal{A}'$, where

$\mathcal{A}' = \{Y \subseteq \mathcal{Y}, |Y| = 1\}$. Similarly, the hypothesis class $\mathcal{H}$ consists of functions $h : \mathcal{X} \to \mathcal{A}$. By definition, there is a bijection between $\mathcal{Y}$ and $\mathcal{A}'$ mapping each $y$ to a corresponding $Y$ such that $y \in Y$. According to this bijection, the loss function $\mathcal{L}$ can be defined on $(\mathcal{A} \times \mathcal{A}')$ such that $\mathcal{L}(h(x), c(x)) = 0$ if and only if $c(x) \subseteq h(x)$. In this way, a unified PAC learning instance $\mathcal{Q} = (\mathcal{X}, \mathcal{Y}, \mathcal{A}, \mathcal{L}, \mathcal{C}, \mathcal{H})$ can be specified as a **L**-list learning instance $(\mathcal{X}, \mathcal{Y}, \mathcal{A}, \mathcal{C}, \mathcal{H})$. Thus, Theorem 4.1 can be applied to this learning setting, together with Theorem 2 and Algorithm 1 in Charikar & Pabbaraju (2023), we have the following corollary.

**Corollary 4.4** (Agnostic $\to$ Realizable (**L**-List Learning)). *If $\mathcal{H}$ has finite **L**-DS dimension $d$, Algorithm 1 is an agnostic list learner with sample complexity:*

$$\mathcal{M}^{\mathrm{AG}}(\epsilon, \delta) = \tilde{\mathcal{O}}\left( \frac{\mathbf{L}^6 d^{3/2} + \log(1/\delta)}{\epsilon^3} \right).$$

### 4.4. Multilabel Learning

With Theorem 4.1 in mind, we can also adapt it to multilabel learning setting where the action space is defined as $\mathcal{A} = \{Y \subseteq \mathcal{Y}, |Y| \leq \mathbf{L}\}$. The concept class $\mathcal{C}$ consists of functions $c : \mathcal{X} \to \mathcal{A}$. Similarly, the hypothesis class $\mathcal{H}$ consists of functions $h : \mathcal{X} \to \mathcal{A}'$, where $\mathcal{A}' = \{Y \subseteq \mathcal{Y}, |Y| = 1\}$. By definition, there is a bijection between $\mathcal{Y}$ and $\mathcal{A}'$ mapping each $y$ to a corresponding $Y$ such that $y \in Y$. According to this bijection, the loss function $\mathcal{L}$ can be defined on $(\mathcal{A} \times \mathcal{A}')$ such that $\mathcal{L}(c(x), h(x)) = 0$ if and only if $h(x) \subseteq c(x)$. In this way, a unified PAC learning instance $\mathcal{Q} = (\mathcal{X}, \mathcal{Y}, \mathcal{A}, \mathcal{L}, \mathcal{C}, \mathcal{H})$ can be specified as a multilabel learning instance $(\mathcal{X}, \mathcal{Y}, \mathcal{A}, \mathcal{C}, \mathcal{H})$, and thus, we can apply Theorem 4.1 to this learning setting.

## 5. Noise Models

In this section, we provide the sample complexity of our multiclass classification reduction algorithm under two noise models: Massart Noise (Massart & Nédélec, 2006) and Tsybakov noise (Tsybakov, 2004). We use $\mathcal{M}_{\mathbb{D}}(\epsilon, \delta)$ to denote the agnostic sample complexity constrained by the distribution $\mathcal{D} \in \mathbb{D}$. Since multiclass learnability can be characterized by DS dimension $d$, which means a hypothesis class $\mathcal{H}$ is PAC learnable if and only if it has finite DS dimension, we restricted our reduction on the hypothesis classes with finite DS dimension. For the realizable learner **A**, we use the one-inclusion graph algorithm introduced by Brukhim et al. (2022).

Before present our result, we first give the definition of these two noise models.

**Definition 5.1** (Massart Noise). *For $\Delta \in (0, 1)$, define $\mathrm{MN}(\Delta)$ as the collection of joint distributions $P_{XY}$ over*

$\mathcal{X} \times \mathcal{Y}$ such that $f^* \in \mathcal{C}$ and

$$P(y = f^*(x)|x) - \max_{y' \neq f^*(x)} P(y = y'|x) \geq \Delta \quad (3)$$

*holds almost surely, where $f^*$ is the Bayes optimal classifier.*

**Definition 5.2** (Tsybakov Noise). *For $a \in [1, \infty)$ and $\alpha \in (0, 1)$, define $\mathrm{TN}(a, \alpha)$ as the collection of joint distributions $P_{XY}$ over $\mathcal{X} \times \mathcal{Y}$ such that $f^* \in \mathcal{C}$ and $\forall \gamma > 0$,*

$$P_X\left( x : P(y = f^*(x)|x) - \max_{y' \neq f^*(x)} P(y = y'|x) \leq \gamma \right)$$
$$\leq a'\gamma^{\alpha/(1-\alpha)},$$
$$(4)$$

*where $a' = (1 - \alpha)(2\alpha)^{\alpha/(1-\alpha)} a^{1/(1-\alpha)}$ and $f^*$ is the Bayes optimal classifier.*

The following theorem shows the reduction from Massart noise to realizable.

**Theorem 5.3** (Massart $\to$ Realizable (Multiclass Learning)). *If $\mathcal{H}$ has finite DS dimension $d$, Algorithm 1 is an agnostic multiclass learner with sample complexity :*

$$\mathcal{M}_{\mathrm{MN}(\Delta)}(\epsilon, \delta) = \tilde{\mathcal{O}}\left( \frac{d^{3/2} + \log(1/\delta)}{\Delta \epsilon^2} \right).$$

Before providing the proof of Theorem 5.3, we state a useful lemma that will be used in the proof.

**Lemma 5.4.** *For any distribution $\mathcal{D}$ over $\mathcal{X} \times \mathcal{Y}$, with probability at least $1 - \delta$, we have:*

$$L_{\mathcal{D}}(\hat{h}) - L_{\mathcal{D}}(c^*) - \epsilon/2 \leq \mathfrak{c}\log(|\mathcal{H}_V|/\delta)/|T|$$
$$+ \mathfrak{c}\sqrt{\left(P(\hat{h}(x) \neq c^*(x)) + \epsilon/2\right)\log(|\mathcal{H}_V|/\delta)/|T|},$$

*where $\mathcal{H}_V$ is the hypothesis class generated in step 2 of Algorithm 1, and $\hat{h}$ is the hypothesis output by ERM in step 3 of Algorithm 1, and $c^* \in \mathcal{C}$ is the Bayes optimal classifier, and $\mathfrak{c}$ is some positive constant.*

*Proof of Lemma 5.4.* By uniform Bernstein inequality, for all $h, h' \in \mathcal{H}_V$, and distribution $\mathcal{D}$ over $(\mathcal{X} \times \mathcal{Y})$, with probability at least $1 - \delta/3$ we have:

$$L_{\mathcal{D}}(h) - L_{\mathcal{D}}(h') \leq \mathfrak{c}\frac{\log(|\mathcal{H}_V|/\delta)}{|T|}$$
$$+ \mathfrak{c}\sqrt{P(h(x) \neq h'(x), y \in \{h(x), h'(x)\})\frac{\log(|\mathcal{H}_V|/\delta)}{|T|}},$$
$$(5)$$

For $\hat{h}, h_{S^*} \in \mathcal{H}$ and $c^* \in \mathcal{C}$, we have the following inequality:

$$P(h_{S^*} \neq \hat{h}(x), y \in \{\hat{h}, h_{S^*}(x)\})$$
$$\leq P(\hat{h}(x) \neq c^*(x), y \in \{\hat{h}(x), c^*(x)\}) \quad (6)$$
$$+ P(h_{S^*}(x) \neq c^*(x), y \in \{h_{S^*}(x), c^*(x)\}).$$

We can furthur upper bound $P(h_{S^*}(x) \neq c^*(x), y \in \{h_{S^*}(x), c^*(x)\})$ by $\epsilon/2$ due to the construction of $h_{S^*}$ in Lemma 4.2. Then combining Equation (5), Equation (6) and Lemma 4.2, we can get the desired conclusion. $\square$

With the above lemma in mind, we start the proof of Theorem 5.3.

*Proof of Theorem 5.3.* For all distribution $\mathcal{D} \in \mathrm{MN}(\Delta)$ and $h \in \mathcal{Y}^{\mathcal{X}}$, with probability 1, we have

$$L_D(h) - L_D(c^*) \geq \Delta P(h(x) \neq c^*(x)). \qquad (7)$$

Combining Lemma 5.4 and Equation (7), with probability at least $1 - \delta$, we have

$$
\begin{aligned}
L_{\mathcal{D}}(\hat{h}) - L_{\mathcal{D}}(c^*) - \epsilon/2 &\leq \mathfrak{c}\frac{\log(|\mathcal{H}_V|/\delta)}{|T|} \\
&+ \mathfrak{c}\sqrt{\left(\frac{1}{\Delta}(L_{\mathcal{D}}(\hat{h}) - L_{\mathcal{D}}(c^*)) + \epsilon/2\right)\frac{\log(|\mathcal{H}_V|/\delta)}{|T|}},
\end{aligned}
\qquad (8)
$$

where $\mathfrak{c}$ is some positive constant. Transferring Equation (8) such that the term $L_{\mathcal{D}}(\hat{h}) - L_{\mathcal{D}}(c^*)$ only appears in the left side of the inequality, with probability at least $1 - \delta$, we have

$$L_{\mathcal{D}}(\hat{h}) - L_{\mathcal{D}}(c^*) \leq \mathfrak{c}'\frac{1}{\Delta}\frac{\log(|\mathcal{H}_V|/\delta)}{|T|} + \frac{\epsilon}{2}, \qquad (9)$$

where $\mathfrak{c}'$ is some positive constant. Further upper bound Equation (9) with $\epsilon$ and solve for $|T|$, we have when $|T| \geq \mathfrak{c}'\log(|\mathcal{H}_V|/\delta)/\Delta\epsilon$, with probability at least $1 - \delta$, the output of Algorithm 1 is a desired learner under Massart noise. The sample complexity follows from noting that $|\mathcal{H}_V|$ is at most $2^{|V|}$ with $|V| = \mathcal{O}(\max_{\mu\in[\epsilon/2,1]}\{\mathcal{M}^{\mathrm{RE}}(\epsilon/(2\mu), \delta/3)/\mu\})$ due to Lemma 4.2. $\square$

The following theorem shows the reduction from Tsybakov noise to realizable.

**Theorem 5.5** (Tsybakov $\rightarrow$ Realizable (Multiclass Learning))**.** *If $\mathcal{H}$ has finite DS dimension $d$, Algorithm 1 is an agnostic multiclass learner with sample complexity:*

$$\mathcal{M}_{\mathrm{TN}(a,\alpha)}(\epsilon, \delta) = \tilde{\mathcal{O}}\left(\frac{a(d^{3/2} + \log(1/\delta))}{\epsilon^{3-\alpha}}\right).$$

*proof of Theorem 5.5.* For all distribution $\mathcal{D} \in \mathrm{TN}(a, \alpha)$ and $h \in \mathcal{Y}^{\mathcal{X}}$, we have

$$P(h(x) \neq c^*(x)) \leq a(L_D(h) - L_D(c^*))^\alpha, \qquad (10)$$

which also refers to Bernstein class condition (Hanneke & Yang, 2015). Combining Lemma 5.4 and Equation (10),

with probability at least $1 - \delta$, we have

$$
\begin{aligned}
L_{\mathcal{D}}(\hat{h}) - L_{\mathcal{D}}(c^*) - \epsilon/2 &\leq \mathfrak{c}\frac{\log(|\mathcal{H}_V|/\delta)}{|T|} \\
&+ \mathfrak{c}\sqrt{\left(a(L_{\mathcal{D}}(\hat{h}) - L_{\mathcal{D}}(c^*))^\alpha + \epsilon/2\right)\frac{\log(|\mathcal{H}_V|/\delta)}{|T|}},
\end{aligned}
\qquad (11)
$$

where $\mathfrak{c}$ is some positive constant. For $|T| \geq \mathfrak{c}''\log(|\mathcal{H}_V|/\delta)/\epsilon^{2-\alpha}$, with probability at least $1 - \delta$, Equation (11) holds with $L_{\mathcal{D}}(\hat{h}) - L_{\mathcal{D}}(c^*) \leq \epsilon$, where $\mathfrak{c}''$ is some positive constant. In this way, we have that the output of Algorithm 1 is a desired learner under Tsybakov noise. The sample complexity follows from noting that $|\mathcal{H}_V|$ is at most $2^{|V|}$ with $|V| = \mathcal{O}(\max_{\mu\in[\epsilon/2,1]}\{\mathcal{M}^{\mathrm{RE}}(\epsilon/(2\mu), \delta/3)/\mu\})$ due to Lemma 4.2. $\square$

# 6. Lower Bound

In this section, we demonstrate that although our sample complexity upper bound includes an additional $1/\epsilon$ term compared to the upper bound obtained via a compression scheme, this extra factor is sometimes unavoidable due to the representation-preserving property maintained by our algorithm. Formally, we state the following theorem:

**Theorem 6.1.** *For every $n, m \in \mathbb{N}$, there exist an example of an instance of the Unified PAC Learning framework $\mathcal{Q} = (\mathcal{X}, \mathcal{Y}, \mathcal{A}, \mathcal{L}, \mathcal{C}, \mathcal{H})$ such that $|\mathcal{X}| = n$, $|\mathcal{Y}| = |\mathcal{A}| = |\mathcal{C}| = |\mathcal{H}| = m$ for which we have:*

$$\forall\epsilon, \delta \in (0, 1), \frac{\mathcal{M}^{RE}(\epsilon, \delta)}{\mathcal{M}^{AG}(\epsilon, \delta)} \in \Omega(1/\epsilon^2),$$

*where $\mathcal{M}^{RE}(\epsilon, \delta)$ and $\mathcal{M}^{AG}(\epsilon, \delta)$ are optimal sample complexity of the realizable PAC learning and the agnostic PAC learning, accordingly.*

*Proof.* Let $\mathcal{X} = \{1, 2, \ldots, n\}$. Also, let $\mathcal{Y} = \mathcal{A} = \{1, 2, \ldots, m\}$. In addition, let $\mathcal{C} = \mathcal{H} = \{c_y \mid c_y : \mathcal{X} \rightarrow \{y\}, y \in \mathcal{Y}\}$. Additionally, define $\mathcal{L}$ as the multiclass classification loss 4.2. Now, observe that $\mathcal{Q}$ is realizable PAC learnable with only one sample. Therefore, we have: $\forall\epsilon, \delta \in (0, 1), \mathcal{M}^{RE}(\epsilon, \delta) \in \mathcal{O}(1)$. Also, we have: $\forall\epsilon, \delta \in (0, 1), \mathcal{M}^{AG}(\epsilon, \delta) \in \Omega(1/\epsilon^2)$. This follows from the standard technique based on the slightly unbiased coin; for instance, see (Daniely et al., 2011). As a result, we have: $\forall\epsilon, \delta \in (0, 1), \mathcal{M}^{RE}(\epsilon, \delta)/\mathcal{M}^{AG}(\epsilon, \delta) \in \Omega(1/\epsilon^2)$. We note that it is possible to construct a similar example using partial concept classes B.1. This finishes the proof. $\square$

*Remark* 6.2. Indeed, it is preferable to provide an example of an instance of the Unified PAC Learning framework where the aforementioned result holds true but we also have an $\Omega(1/\epsilon)$ lower bound on the optimal sample complexity of the realizable PAC learning for every $\epsilon, \delta \in (0, 1)$ as the

parameters. However, this claim holds if and only if there is a sublinear proper sample compression scheme for a given hypothesis class, which is an interesting and important topic to explore. Actually, if we have a proper sample compression scheme of size $k(n) = \mathcal{O}\left(n^{1/3}/\log n\right)$, then we can get an agnostic proper sample compression scheme with the same size $k(n)$. An agnostic proper sample compression scheme of size $k(n)$ implies a proper agnostic learner with error rate

$$\epsilon(n, \delta) = \mathcal{O}\left(\sqrt{\frac{k(n)\log\frac{n}{k(n)} + k(n) + \log\frac{1}{\delta}}{n}}\right).$$

In other words, the sample complexity of this agnostic learner would be $\mathcal{M}^{AG}(\epsilon, \delta) = \mathcal{O}(1/\epsilon^3)$, which matches the sample complexity upper bound of our algorithm. However, even in the binary case, the existence of such a proper sample compression scheme remains unproven. We leave this as an open question for further exploration.

## 7. Conclusion

In this work, we established a *simple* and *elegant* agnostic to realizable reduction within the *Unified PAC Learning framework*. In particular, our result leads to the first *representation preserving* agnostic to realizable reduction within the multiclass PAC learning framework, especially when the number of labels can be unbound. Also, we explored representation preserving reductions in the realizable setting for two noise models in the multiclass PAC learning framework. In addition, we showed similar results for the Unified PAC Learning framework with *partial concept classes*. We believe that our techniques could have applications in future studies in the field of statistical learning theory.

## Impact Statement

This paper presents work whose goal is to advance the field of Statistical Learning Theory. There are many potential societal consequences of our work, none of which we feel must be specifically highlighted here.

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

## A. Failure of Hopkins et al. (2022) in Infinite-Label Multiclass Learning

We address it by noting the Hopkins et al. reduction fails for the well-studied "stars and sets" concept class, namely Example 4.1 of Daniely et al. (2011). Specifically, consider $\mathcal{X} = [0, 1]$, denote by $\mathcal{F}(\mathcal{X})$ the collection of all finite subsets $A \subseteq \mathcal{X}$. Let the label space $\mathcal{Y} = \mathcal{F}(\mathcal{X}) \bigcup \{\star\}$, where $\star$ is a special label (not to be confused with "$*$" from the partial concept classes section). For every $A \subseteq \mathcal{X}$, define $f_A : \mathcal{X} \to \mathcal{Y}$ as follows:

$$f_A(x) = \begin{cases} \star & \text{if } x \in A \\ A & \text{otherwise.} \end{cases}$$

Let the concept and hypothesis class be $\mathcal{C} = \mathcal{H} = \{f_A : A \in \mathcal{F}(\mathcal{X}) \cup \mathcal{X}\}$. Then, there is a realizable learner $\mathcal{A}_{\text{good}}$ that returns $f_{\mathcal{X}}$ unless a label $A \in \mathcal{F}(\mathcal{X})$ appeared in the sample, in which case it returns $f_A$. Let $\mathcal{D}$ have marginal on $\mathcal{X}$ the uniform distribution over $\mathcal{X}$, and let the labels be always $\star$ (i.e., realizable with target $f_{\mathcal{X}}$). Now, consider the algorithm of Hopkins et al. with this scenario and realizable learner $\mathcal{A}_{\text{good}}$. Let the unlabeled dataset be $S_U = \{x_1, x_2, \cdots, x_n\}$, and the labeled dataset be $S_L = \{(x_{n+1}, \star), (x_{n+2}, \star), \cdots, (x_{n+m}, \star)\}$, and with probability one these $x$'s are all distinct. Denote $A = \{x_{n+1}, x_{n+2}, \cdots, x_{n+m}\} \in \mathcal{F}(\mathcal{X})$. Then their algorithm would run $\mathcal{A}_{\text{good}}$ on all realizable labelings of $S_U$; in particular, one of these is $(S_U, f_A(S_U)) = \{(x_1, A), \ldots, (x_n, A)\}$, and the output hypothesis of $\mathcal{A}_{\text{good}}(S_U, f_A(S_U))$ would be $f_A$. By the definition of $f_A$, we know that the empirical error of $f_A$ on $S_L$ is 0. Their algorithm then outputs any ERM on $S_L$ from these functions produced by $\mathcal{A}_{\text{good}}$, which means their algorithm can output $f_A$. However, the true error rate of $f_A$ is 1, while the best error in the concept class is 0. Thus, their algorithm fails for this concept class (for essentially the same reason ERM fails for this concept class). In contrast, our algorithm returns $f_{\mathcal{X}}$, hence achieves error 0.

## B. Partial Concept Classes

We discuss the reduction for partial PAC learning in this section.

### B.1. Notations

We first develop some necessary notation for partial PAC learning. A partial predictor is a function from $\mathcal{X}$ to $\mathcal{A} \cup \{\star\}$. With this in mind, fix a non-empty set of partial predictors $\mathcal{C}$ as a concept class. Additionally, fix another set of predictors $\mathcal{H}$ as a hypothesis class. The support of a partial predictor $h$ is the set $\text{supp}(h) = h^{-1}(\mathcal{A})$. A loss function $\mathcal{L}$ is a map $\mathcal{L} : \mathcal{A} \cup \{\star\} \times \mathcal{Y} \to \{0, 1\}$, with $\mathcal{L}(\star, \cdot) \equiv 0$. Similar to unified PAC learning, a 6-tuple $\mathcal{Q}$ presents an instance of Unified Partial PAC Learning framework. A learning algorithm $\mathbf{A}$ is a mapping from $(\mathcal{X} \times \mathcal{Y})^*$ to a partial predictor $\hat{h}$.

Given the above notations, we make the following definitions for realizability. A probability distribution, $\mathcal{D}$, over the product space of instance space and label space $(\mathcal{X} \times \mathcal{Y})$, is realizable by a partial concept class $\mathcal{C}$ if almost surely (i.e., with probability 1), a sample $S = ((x_i, y_i))_{i=1}^n \sim D^n$ (for any $n \in \mathbb{N}$) is realizable by some partial concept $c \in \mathcal{C}$: that is $\{x_i\}_{i=1}^n \subset \text{supp}(c)$ and $\mathcal{L}(c(x_i), y_i) = 0$ for all $i \leq n$. For a partial concept $c$ and a distribution $D$ on $\mathcal{X} \times \mathcal{Y}$, we define the true risk: $L_{\mathcal{D}}(c) := \mathbb{E}_{(x,y) \sim \mathcal{D}}[\mathcal{L}(c(x), y)]$. The true risk of a hypothesis $h$ can also be defined as $L_{\mathcal{D}}(h) := \mathbb{E}_{(x,y) \sim \mathcal{D}}[\mathcal{L}(h(x), y)]$. Given the above notation and definition, we define Unified Partial PAC learning, which can be easily adapted to a specific learning regime.

**Definition B.1** (Unified Partial PAC Learning). We say that $\mathcal{Q} = (\mathcal{X}, \mathcal{Y}, \mathcal{A}, \mathcal{L}, \mathcal{C}, \mathcal{H})$ is unified partial PAC learnable, if for every $\epsilon, \delta \in (0, 1)$, there exists a finite $\mathcal{M}^{\text{RE}}(\epsilon, \delta) \in \mathbb{N}$ and a learning algorithm $\mathbf{A}$ with $\text{Im}(\mathbf{A}) \subseteq \mathcal{H}$ such that, for every distribution $\mathcal{D}$ on $\mathcal{X} \times \mathcal{Y}$ realizable w.r.t. $\mathcal{C}$, for $S \sim \mathcal{D}^{\mathcal{M}^{\text{RE}}(\epsilon, \delta)}$, with probability at least $1 - \delta$,

$$L_{\mathcal{D}}(\mathbf{A}(S)) \leq \epsilon.$$

The value $\mathcal{M}^{\text{RE}}(\epsilon, \delta)$ is called the sample complexity of $\mathbf{A}$, and the optimal sample complexity[4] is the minimum achievable value of $\mathcal{M}^{\text{RE}}(\epsilon, \delta)$ for every given $\epsilon, \delta$.

For any $n \in \mathbb{N}$ and data sequence $S \in (\mathcal{X} \times \mathcal{Y})^n$, define the empirical risk of any partial concept $c$ as $L_S(c) = \frac{1}{n} \sum_{i=1}^n \mathcal{L}(c(x_i), y_i)$. For a distribution $\mathcal{D}$ on $\mathcal{X} \times \mathcal{Y}$, define the approximation error of $\mathcal{C}$ as $L_{\mathcal{D}}(\mathcal{C}) = \lim_{n \to \infty} \mathbb{E}_{S \sim \mathcal{D}^n}[\min_{c \in \mathcal{C}} L_S(c)]$.

---

[4]For brevity, we refer to the optimal sample complexity as sample complexity.

**Definition B.2** (Unified Agnostic Partial PAC Learning). We say that $\mathcal{Q} = (\mathcal{X}, \mathcal{Y}, \mathcal{A}, \mathcal{L}, \mathcal{C}, \mathcal{H})$ is agnostic unified partial PAC learnable, if for every $\epsilon, \delta \in (0, 1)$, there exists a finite $\mathcal{M}^{\mathrm{AG}}(\epsilon, \delta) \in \mathbb{N}$ and a learning algorithm $\mathbf{A}$ with $\mathrm{Im}(\mathbf{A}) \subseteq \mathcal{H}$ such that, for every distribution $\mathcal{D}$ on $\mathcal{X} \times \mathcal{Y}$, for $S \sim \mathcal{D}^{\mathcal{M}^{\mathrm{AG}}(\epsilon, \delta)}$, with probability at least $1 - \delta$,

$$L_{\mathcal{D}}(\mathbf{A}(S)) \leq L_{\mathcal{D}}(\mathcal{C}) + \epsilon.$$

The value $\mathcal{M}^{\mathrm{AG}}(\epsilon, \delta)$ is called the sample complexity of $\mathbf{A}$, and the optimal sample complexity is the minimum achievable value of $\mathcal{M}^{\mathrm{AG}}(\epsilon, \delta)$ for every given $\epsilon, \delta$.

## B.2. General Reduction

Given the definition above, we provide the unified partial agnostic to realizable reduction in the following theorem.

**Theorem B.3** (Agnostic $\rightarrow$ Realizable (Partial Learning) ). *Let $\mathbf{A}$ be a realizable unified partial learner for a learning instance $\mathcal{Q} = (\mathcal{X}, \mathcal{Y}, \mathcal{A}, \mathcal{L}, \mathcal{C}, \mathcal{H})$ with sample complexity $\mathcal{M}^{\mathrm{RE}}(\epsilon, \delta)$. Then Algorithm 1 is an agnostic unified partial learner for $(\mathcal{X}, \mathcal{Y}, \mathcal{A}, \mathcal{L}, \mathcal{C}, \mathcal{H})$ with sample complexity:*

$$\mathcal{M}^{\mathrm{AG}}(\epsilon, \delta) = \mathcal{O}\left(\frac{\max_{\mu \in [\epsilon/2, 1]} \left\{\frac{\mathcal{M}^{\mathrm{RE}}(\epsilon/2\mu, \delta/3)}{\mu}\right\} + \log(1/\delta)}{\epsilon^2}\right).$$

Theorem B.3 is also the first representation preserving reduction within unified partial PAC learning regime. However, compared to a reduction-to-realizable technique of David et al. (2016), Theorem B.3 incurs an extra $\max_{\mu \in [\epsilon/2, 1]}\{\mathcal{M}^{\mathrm{RE}}(\epsilon/2\mu, \delta/3)/\mu\}$ term, which can scale to $1/\epsilon$. This raises an open question:

Although Theorem B.3 achieves the same sample complexity as Theorem 4.1, the proofs differ significantly, as there is no optimal concept $c^*$ in $\mathcal{C}$ under the framework of agnostic partial PAC learning. Before presenting the proof, we introduce the following notations. Let $\hat{h}$ be the output hypothesis in step 3 of Algorithm 1. For a given sample $T$, let $c^* = \arg\min_{c \in \mathcal{C}} L_T(c)$. According to $c^*$, let $S^* \subset S$ be the largest realizable sample set in $S$, i.e. $S^* = \{(x, y) \in S | \mathcal{L}(c^*(x) = y)\}$. Also, let $h_{S^*} = \mathbf{A}(S^*)$. With these notations in mind, we we now present the proof of Theorem B.3.

*Proof of Theorem B.3.* Leveraging a standard Chernoff bound over $\mathcal{H}_S$, for any $h \in \mathcal{H}_S$, with probability at least $1 - \delta/4$, we have

$$L_D(h) \leq L_T(h) + \mathfrak{c}_1 \sqrt{\log(|\mathcal{H}_S|/\delta)/|T|},$$

for some positive constant $\mathfrak{c}_1$. Specially, for the output hypothesis $\hat{h}$ in step 3 of Algorithm 1, with probability at least $1 - \delta/4$, we have

$$L_D(\hat{h}) \leq L_T(\hat{h}) + \mathfrak{c}_1 \sqrt{\log(|\mathcal{H}_S|/\delta)/|T|}.$$

By the construction of $\hat{h}$, we have $L_T(\hat{h}) \leq L_T(h)$ for all $h \in \mathcal{H}_S$. Since $h_{S^*} \in \mathcal{H}_S$, we have $L_T(\hat{h}) \leq L_T(h_{S^*})$, which futhur leads to

$$L_D(\hat{h}) \leq L_T(h_{S^*}) + \mathfrak{c}_1 \sqrt{\log(|\mathcal{H}_S|/\delta)/|T|} \tag{12}$$

holds with probability at least $1 - \delta/4$. On the other hand, since function $L_T(h)$ is bounded noise with constant bound $1/|T|$, leveraging McDiarmid's inequality, with probability at least $1 - \delta/4$, we have

$$L_T(c^*) \leq \mathbb{E}[L_T(c^*)] + \mathfrak{c}_2 \sqrt{\log(1/\delta)/|T|},$$

for some postive constant $\mathfrak{c}_s$. Since $\mathbb{E}[L_T(c^*)]$ is non decreasing in the size of sample $T$, (see Lemma 39 in Alon et al. (2022)) we have $\mathbb{E}[L_T(c^*)] \leq L_D(\mathcal{C})$ for any $T$ sampled from distribution $D$. Thus, with probability at least $1 - \delta/4$, we have

$$L_T(c^*) \leq L_D(\mathcal{C}) + \mathfrak{c}_2 \sqrt{\log(1/\delta)/|T|}, \tag{13}$$

for some positive constant $\mathfrak{c}_2$. If we can further bound the difference $L_T(h_{S^*}) - L_T(c^*)$ with high probability, combining Equation (12) and (13), we can get the desired conclusion. A natural idea is to divide $T$ into two parts: $T^*$, the realizable part according to $c^*$, and $\bar{T}^*$, the complementary of $T^*$. Then, we can decompose $L_T(c^*)$ as

$$L_T(c^*) = \mu^* L_{T^*}(c^*) + (1 - \mu^*) L_{\bar{T}^*}(c^*) = 1 - \mu^*, \tag{14}$$

where $\mu^* = |T^*|/|T|$ denote the portion of realizable samples of $c^*$ in $T$. Similarly, we can decompose $L_T(h_{S^*})$ as

$$
\begin{aligned}
L_T(h_{S^*}) &= \mu^* L_{T^*}(h_{S^*}) + (1 - \mu^*) L_{\bar{T}^*}(h_{S^*}) \\
&\leq 1 - \mu^* + \mu^* L_{T^*}(h_{S^*}) \\
&= L_T(c^*) + \mu^* L_{T^*}(h_{S^*}),
\end{aligned}
\tag{15}
$$

where the inequality comes from $L_{\bar{T}^*}(h_{S^*}) \leq 1$, and the last equality is due to Equation (14). Since both $S^*$ and $T^*$ are realizable by $c^*$, as long as $|S^*| \geq \mathcal{M}^{\mathrm{RE}}(\epsilon/2\mu^*, \delta/4)$, with probability at least $1 - \delta/4$, we have $L_{T^*}(h^{S^*}) \leq \epsilon/2$. Leveraging a standard Chernoff bound, when $|S| = \mathfrak{c}_3 \max_{\mu \in [\epsilon/2, 1]}\{\mathcal{M}^{\mathrm{RE}}(\epsilon/2\mu, \delta/4)/\mu\}$, with probability at least $1 - \delta/4$, we have $|S^*| \geq \mathcal{M}^{\mathrm{RE}}(\epsilon/2\mu^*, \delta/4)$. Choosing $|T| = \mathcal{O}(\log(|\mathcal{H}_S|/\delta)/\epsilon^2)$, by a union bound across the whole process, we have, with probability at least $1 - \delta$,

$$
L_D(\hat{h}) \leq L_D(\mathcal{C}) + \epsilon.
$$

In this way, we prove Algorithm 1 is an agnostic partial unified PAC leaner. The sample complexity comes from $|\mathcal{H}_S| = 2^{|S|}$ with $|S| = \mathfrak{c}_3 \max_{\mu \in [\epsilon/2, 1]}\{\mathcal{M}^{\mathrm{RE}}(\epsilon/2\mu, \delta/4)/\mu\}$ for some positive constant $\mathfrak{c}_3$. $\qquad\square$

## B.3. Multiclass Learning

With Theorem B.3 in mind, we can readily adapt it to multiclass learning setting with the action space $\mathcal{A} = \mathcal{Y} \cup \{\star\}$, and $\mathcal{L}$ represents the classification error. In this way, a unified PAC learning instance $\mathcal{Q} = (\mathcal{X}, \mathcal{Y}, \mathcal{A}, \mathcal{L}, \mathcal{C}, \mathcal{H})$ can be specified as a multiclass learning instance $(\mathcal{X}, \mathcal{Y}, \mathcal{C}, \mathcal{H})$.

**Corollary B.4** (Agnostic $\to$ Realizable (Partial Multiclass Learning))**.** *Let $\mathbf{A}$ be a realizable partial multiclass learner for a learning instance $(\mathcal{X}, \mathcal{Y}, \mathcal{C}, \mathcal{H})$ with sample complexity $\mathcal{M}^{\mathrm{RE}}(\epsilon, \delta)$. Then Algorithm 1 is an agnostic partial multiclass learner for $(\mathcal{X}, \mathcal{Y}, \mathcal{C}, \mathcal{H})$ with sample complexity:*

$$
\mathcal{M}^{\mathrm{AG}}(\epsilon, \delta) = \mathcal{O}\left( \frac{\max_{\mu \in [\epsilon/2, 1]}\left\{ \frac{\mathcal{M}^{\mathrm{RE}}(\epsilon/2\mu, \delta/3)}{\mu} \right\} + \log(1/\delta)}{\epsilon^2} \right),
$$

*Moreover, if $\mathcal{H}$ has finite DS dimension $d$, Algorithm 1 only needs:*

$$
\mathcal{M}^{\mathrm{AG}}(\epsilon, \delta) = \mathcal{O}\left( \frac{d^{3/2} + \log(1/\delta)}{\epsilon^3} \right)
$$

*labeled samples.*

## B.4. List Learning

Building on Theorem B.3, We can adapt it to $\mathbf{L}$-list learning setting with the action space $\mathcal{A} = \{Y \subseteq \mathcal{Y}, |Y| \leq \mathbf{L}\}$. The concept class $\mathcal{C}$ consists of functions $c : \mathcal{X} \to \mathcal{A}' \cup \{\star\}$, where $\mathcal{A}' = \{Y \subseteq \mathcal{Y}, |Y| = 1\}$. Similarly, the hypothesis class $\mathcal{H}$ consists of functions $h : \mathcal{X} \to \mathcal{A} \cup \{\star\}$. By definition, there is a bijection between $\mathcal{Y}$ and $\mathcal{A}'$ mapping each $y$ to a corresponding $Y$ such that $y \in Y$. According to this bijection, the loss function $\mathcal{L}$ can be defined on $(\mathcal{A} \cup \{\star\} \times \mathcal{A}' \cup \{\star\})$ such that $\mathcal{L}(h(x), c(x)) = 0$ if and only if $c(x) \subseteq h(x)$. In this way, a unified PAC learning instance $\mathcal{Q} = (\mathcal{X}, \mathcal{Y}, \mathcal{A}, \mathcal{L}, \mathcal{C}, \mathcal{H})$ can be specified as a $\mathbf{L}$-list learning instance $(\mathcal{X}, \mathcal{Y}, \mathcal{A}, \mathcal{C}, \mathcal{H})$.

**Corollary B.5** (Agnostic $\to$ Realizable (Partial List Learning))**.** *Let $\mathcal{A}$ be a realizable partial list learner for a learning instance $(\mathcal{X}, \mathcal{Y}, \mathcal{A}, \mathcal{C}, \mathcal{H})$ with sample complexity $\mathcal{M}^{\mathrm{RE}}(\epsilon, \delta)$. Then Algorithm 1 is an agnostic partial list learner for $(\mathcal{X}, \mathcal{Y}, \mathcal{A}, \mathcal{C}, \mathcal{H})$ with sample complexity:*

$$
\mathcal{M}^{\mathrm{AG}}(\epsilon, \delta) = \mathcal{O}\left( \frac{\max_{\mu \in [\epsilon/2, 1]}\left\{ \frac{\mathcal{M}^{\mathrm{RE}}(\epsilon/2\mu, \delta/3)}{\mu} \right\} + \log(1/\delta)}{\epsilon^2} \right).
$$

*Moreover, if $\mathcal{H}$ has finite $\mathbf{L}$-DS dimension $d$, Algorithm 1 needs only:*

$$
\mathcal{M}^{\mathrm{AG}}(\epsilon, \delta) = \mathcal{O}\left( \frac{\mathbf{L}^6 d^{3/2} + \log(1/\delta)}{\epsilon^3} \right)
$$

*labeled samples.*

### B.5. Multilabel Learning

We can also adapt Theorem B.3 to multilabel learning setting where the action space is defined as $\mathcal{A} = \{Y \subseteq \mathcal{Y}, |Y| \leq \mathbf{L}\}$. The concept class $\mathcal{C}$ consists of functions $c : \mathcal{X} \to \mathcal{A}$. Similarly, the hypothesis class $\mathcal{H}$ consists of functions $h : \mathcal{X} \to \mathcal{A}'$, where $\mathcal{A}' = \{Y \subseteq \mathcal{Y}, |Y| = 1\}$. By definition, there is a bijection between $\mathcal{Y}$ and $\mathcal{A}'$ mapping each $y$ to a corresponding $Y$ such that $y \in Y$. According to this bijection, the loss function $\mathcal{L}$ can be defined on $(\mathcal{A} \cup \{\star\} \times \mathcal{A}' \cup \{\star\})$ such that $\mathcal{L}(c(x), h(x)) = 0$ if and only if $h(x) \subseteq c(x)$. In this way, a unified PAC learning instance $\mathcal{Q} = (\mathcal{X}, \mathcal{Y}, \mathcal{A}, \mathcal{L}, \mathcal{C}, \mathcal{H})$ can be specified as a multilabel learning instance $(\mathcal{X}, \mathcal{Y}, \mathcal{A}, \mathcal{C}, \mathcal{H})$.

**Corollary B.6** (Agnostic $\to$ Realizable (Partial Multilabel Learning)). *Let $\mathcal{A}$ be a realizable partial multilabel learner for $(\mathcal{X}, \mathcal{Y}, \mathcal{A}, \mathcal{C}, \mathcal{H})$ with sample complexity $\mathcal{M}^{\mathrm{RE}}(\epsilon, \delta)$. Then Algorithm 1 is an agnostic partial multilabel learner for $(\mathcal{X}, \mathcal{Y}, \mathcal{A}, \mathcal{C}, \mathcal{H})$ with sample complexity:*

$$\mathcal{M}^{\mathrm{AG}}(\epsilon, \delta) = \mathcal{O}\left( \frac{\max_{\mu \in [\epsilon/2, 1]} \left\{ \frac{\mathcal{M}^{\mathrm{RE}}(\epsilon/2\mu, \delta/3)}{\mu} \right\} + \log(1/\delta)}{\epsilon^2} \right).$$

## C. Reduction via Compression Scheme

For completeness, we state the sample complexity of reduction via compression method in this section.

The following theorem presents the sample complexity of the agnostic to realizable reduction achieved through a compression scheme.

**Theorem C.1** (Agnostic$\to$ Realizable (Compression)). *If $\mathcal{C}$ is unified PAC learnable with sample complexity $\mathcal{M}^{\mathrm{RE}}(\epsilon, \delta)$, then it is unified agnostic PAC learnable with sample complexity:*

$$\mathcal{M}^{\mathrm{AG}}(\epsilon, \delta) = \mathcal{O}\left( \frac{\mathcal{M}^{\mathrm{RE}}(1/3, 1/3) \log^2(1/\epsilon) + \log(1/\delta)}{\epsilon^2} \right).$$

*Proof of Theorem C.1.* The proof follows directly from Theorem 3.1 and Theorem 3.2 in David et al. (2016). $\qquad\square$

The following theorem presents the sample complexity of the Massart noise to realizable reduction achieved through a compression scheme.

**Theorem C.2** (Massart$\to$ Realizable (Compression)). *If a multiclass concept class $\mathcal{C}$ is learnable with sample complexity $\mathcal{M}^{\mathrm{RE}}(\epsilon, \delta)$, then it is also learnable under Massart noise with sample complexity:*

$$\mathcal{M}_{\mathrm{MN}(\Delta)}(\epsilon, \delta) = \mathcal{O}\left( \frac{\mathcal{M}^{\mathrm{RE}}(1/3, 1/3) \log^2(\Delta/\epsilon) + \log(1/\delta)}{\Delta\epsilon} \right).$$

*Proof of Theorem C.2.* According to Theorem 3.2 in David et al. (2016), we have: If $\mathcal{C}$ can be learned with sample complexity $\mathcal{M}^{\mathrm{RE}}(\epsilon, \delta)$, there is a sample compression scheme of size $k(m) = \mathcal{O}(d_0 \cdot \log(m) \log(d_0 \cdot \log(m)))$ where $d_0 = \mathcal{M}^{\mathrm{RE}}(1/3, 1/3)$. Furthermore, a sample compression scheme of size $k(m)$ implies an agnostic sample compression scheme of size $k(m)$. Denote this sample compression scheme by $\mathbf{A}(S)$, where $S$ is the sample with size $|S| = m$. Due to the upper bound on the size of comparison scheme is $k(m)$, the size of the class of compression functions $|\mathcal{H}_c|$ can be upper bounded by

$$\sum_{i}^{k(m)} \binom{m}{i} \leq \left( \frac{em}{k(m)} \right)^{k(m)}. \tag{16}$$

Then a universal Bernstein inequality gives that: for all $\hat{h} \in \mathcal{H}_c$, with probability at least $1 - \delta$, we have

$$\left| (L_S(\hat{h}) - L_S(c^*)) - (L_D(\hat{h}) - L_D(c^*)) \right|$$
$$\leq \mathfrak{c}\sqrt{P_D\left(\hat{h}(x) \neq c^*(x), y \in \{\hat{h}(x), c^*(x)\}\right) \frac{\log(|\mathcal{H}_c|/\delta)}{|S|}} + \mathfrak{c}\frac{\log(|\mathcal{H}_c|/\delta)}{|S|}, \tag{17}$$

where $c^*$ is the Bayesian optimal classifier in $\mathcal{C}$. Specially, for the agnostic compression scheme $\mathbf{A}(S)$, with probability at least $1 - \delta$, we have

$$L_D(\mathbf{A}(S)) - L_D(c^*) \leq \mathfrak{c}\sqrt{P_D\Big(\mathbf{A}(S)(s) \neq c^*(x), y \in \{\mathbf{A}(S)(s), c^*(x)\}\Big)\frac{\log(|\mathcal{H}_c|/\delta)}{|S|}} + \mathfrak{c}\frac{\log(|\mathcal{H}_c|/\delta)}{|S|}, \tag{18}$$

due to the definition of agnostic compression scheme. Combining Equation (18) and Equation (7), with probability at least $1 - \delta$, we have

$$L_D(\mathbf{A}(S)) - L_D(c^*) \leq \mathfrak{c}\sqrt{\frac{1}{\Delta}(L_D(\mathbf{A}(S)) - L_D(c^*))\frac{\log(|\mathcal{H}_c|/\delta)}{|S|}} + \mathfrak{c}\frac{\log(|\mathcal{H}_c|/\delta)}{|S|}. \tag{19}$$

Combining Equation (19) and Equation (16) and solve, we have $L_D(\mathbf{A}(S)) - L_D(c^*) \leq \epsilon$, if

$$|S| = \mathcal{O}\left(\frac{\mathcal{M}^{\mathrm{RE}}(1/3, 1/3)\log^2(\Delta/\epsilon) + \log(1/\delta)}{\Delta\epsilon}\right).$$

In this way, we prove that the compression scheme is a learner under Massart noise and also prove the upper bound on sample complexity. $\qquad\square$

The following theorem presents the sample complexity of the Tsybakov noise to realizable reduction achieved through a compression scheme.

**Theorem C.3** (Tsybakov→ Realizable (Compression)). *If a multiclass concept class $\mathcal{C}$ is learnable with sample complexity $\mathcal{M}^{\mathrm{RE}}(\epsilon, \delta)$, then it is also learnable under Tsybakov noise with sample complexity:*

$$\mathcal{M}_{\mathrm{TN}(a,\alpha)}(\epsilon, \delta) = \mathcal{O}\left(\frac{a(\mathcal{M}^{\mathrm{RE}}(1/3, 1/3)\log^{2\alpha}(1/\epsilon) + \log(1/\delta))}{\epsilon^{2-\alpha}}\right).$$

*Proof of Theorem C.3.* The proof closely mirrors the argument in Theorem C.2, with the key difference being the substitution of the Massart noise condition (i.e. Equation (7)) by the Tsybakov noise condition (i.e. Equation (10)) in Equation (19). $\qquad\square$

# D. Concentration Inequalities

In this section, we list the concentration inequalities used in our paper without providing proofs.

**Theorem D.1** (Generic Chernoff Bound). *For a random variable $X$, and a positive number $t$, and any $a$, we have*

$$P(X \geq a) \leq M(t)\exp(-ta),$$

*where $M(t) = \mathbb{E}[\exp(tX)]$ is the moment generating function of $X$.*

**Theorem D.2** (Uniform Bernstein Inequality). *For all $h, h' \in \mathcal{H}$ and distribution $D$ over $\mathcal{X} \times \mathcal{Y}$, with probability at least $1 - \delta$, we have*

$$|(L_S(h) - L_S(h')) - (L_D(h) - L_D(h'))|$$
$$\leq \mathfrak{c}\sqrt{P_D(h(x) \neq h'(x), y \in \{h(x), h'(x)\})\frac{\log(|\mathcal{H}|/\delta)}{|S|}} + \mathfrak{c}\frac{\log(|\mathcal{H}|/\delta)}{|S|},$$

*where $\mathcal{H}$ is a multiclass hypothesis class, and $\mathfrak{c}$ is some positive constant.*

**Theorem D.3** (McDiarmid's Inequality). *Let $f : \mathcal{X}_1 \times \mathcal{X}_2 \times \cdots \times \mathcal{X}_n \to \mathbb{R}$ satisfy the bounded differences property with bounds $c_1, c_2, \ldots, c_n$. Consider independent random variables $X_1, X_2, \ldots, X_n$ where $X_i \in \mathcal{X}_i$ for all $i$. Then, for any $\epsilon > 0$,*

$$P(f(X_1, X_2, \ldots, X_n) - \mathbb{E}[f(X_1, X_2, \ldots, X_n)] \geq \epsilon) \leq \exp\left(\frac{-2\epsilon^2}{\sum_{i=1}^{n} c_i^2}\right),$$

$$P(f(X_1, X_2, \ldots, X_n) - \mathbb{E}[f(X_1, X_2, \ldots, X_n)] \leq \epsilon) \leq \exp\left(\frac{-2\epsilon^2}{\sum_{i=1}^{n} c_i^2}\right).$$

