# OpenReview forum: "Representation Preserving Multiclass Agnostic to Realizable Reduction"
_ICML.cc/2025/Conference — ICML 2025 poster_

### Official Review · Reviewer_ajtt · 2025-02-20

**Overall Recommendation:** 3

**Summary:**

In the PAC learning model, one is trying to learn a function f over a distribution D. One is given samples and attempts to return a hypothesis h so that Pr_{x ~ D}(f(x) neq h(x)) is small. In the realizable setting, one obtains samples of the form (x,f(x)) where f is guaranteed to be in some function class C and x ~ D. In the agnostic model, one obtains samples of the form (x,y) from some arbitrary distribution (with marginal x ~ D) and tries to output a hypothesis h where Pr(h(x) neq y) is competitive with the best Pr(f(x) neq y) over the best f taken from the class C.

There are many variations of this setting where the existence of a learner in the realizable setting implies the existence of an agnostic learner. [Hopkins et al '22] found a relatively simple reduction that can be used to show in a fairly wide variety of settings that a learning algorithm in the realizable setting can be used to construct an agnostic learner. Their primary focus was on the distribution-family model where the x-marginal D is taken from a known but arbitrary family of distributions. In this context, their construction provided the first known realizable to agnostic reduction but only when the space of values of y was finite (as otherwise the reduction is false without extra assumptions).

This paper shows that a simple modification of the reduction of [Hopkins et al '22] can be used when the space of y's is infinite in the distribution-free model (i.e where D can be taken to be *any* distribution over the domain X). In fact, they prove this in a slightly more general context that includes list-decodable and multi-valued learning problems.

While many of these reductions were already known in [David et al., 2016], the resulting reduction in this paper is substantially simpler and guarantees that the returned hypothesis of the agnostic learner comes from the same class of functions returned by the realizable learner (albeit this is all at the cost of somewhat worse sample complexity than [David et al., 2016]).

**Claims And Evidence:**

Yes. They are supported by mathematical proof.

**Essential References Not Discussed:**

Not that I know of.

**Experimental Designs Or Analyses:**

N/A

**Methods And Evaluation Criteria:**

N/A

**Other Comments Or Suggestions:**

Unless I am misunderstanding the definitions (your definition of partial learners is hard to follow), I believe that Theorem A.3 is wrong.

In particular, let X={a,b}, Y=A={0,1}, L(y,z) = 0 if y=* or y=z and 1 otherwise. C=H consists of two functions:
f(a) = 0, f(b) = *
g(a) = g(b) = 1

Then a 1-sample realizable learning algorithm takes (x,y) and returns g if x = b or y = 1 and f otherwise. This has error 0 since if the true function is f, then D must be supported on (a,0).

However, consider the agnostic learning problem where D is supported on (b,0). Then Algorithm 1 running on this produces only the hypothesis g, which has error 1.

**Other Strengths And Weaknesses:**

While the main result of this paper is nice, the technique is a relatively simple extension of [Hopkins et al] and the headline result was basically already known by [David et al], and I do not think that this result is central enough for this simplified proof to be publishable on its own.

The discussion of the reduced sample complexity in the case of Massart noise was cute, but not enough to save the paper.

**Questions For Authors:**

None

**Relation To Broader Scientific Literature:**

This work expands on the ideas of [Hopkins et al] and produces results similar to those of [David et al] but with some advantages and disadvantages over that work.

**Theoretical Claims:**

Although I didn't read proofs in detail, I was able to convince myself that the results in the main body could all be proved using techniques similar to those mentioned. Although the full proofs may contain small errors, they should be easily fixable.

On the other hand, I believe that Theorem A.3 is wrong.

---

> ### Author Rebuttal · Authors · 2025-04-01
>
> We thank the reviewer for dedicating their time to assess our work. Below, we address the comments provided by the reviewer.
>
> **1. The significance of our work:** We establish the first representation-preserving reduction from agnostic to realizable learning for multiclass classification with an infinite label space, in addition to various other learning settings. Notably, this type of reduction was not presented in the work of David, Moran, and Yehudayoff (2016). This is because their approach relied on a compression-based reduction using boosting, which increases the complexity of the agnostic learner’s hypothesis class compared to that of the realizable learner. In contrast, our algorithm is representation-preserving, meaning the agnostic learner’s hypothesis class remains the same as that of the realizable learner. Thus, our algorithm and results are fundamentally different from theirs.  In particular, ours is the first proof that a class $\mathcal{C}$ is agnostically learnable with hypothesis class $\mathcal{H}$ if and only if $\mathcal{C}$ is realizably learnable with hypothesis class $\mathcal{H}$.
>
> **2. The novelty of our work:** Of course we take inspiration from the work of Hopkins et al., but the key ingredient in our algorithm is novel. In particular, the algorithm of Hopkins et al. runs a realizable learner on an unlabeled sample with all possible labelings, which can only apply to learning settings with a finite effective label space. However, our algorithm runs a realizable learner on all the subsets of a labeled sample, which can handle learning problems with an infinite label space, i.e. multiclass PAC learning with an infinite label space.  Moreover, the analysis of the algorithm is also different, as we must account for the variable size of the optimal subset present in this initial labeled data set. Therefore, we designed a novel algorithm with crucial technical differences compared to previous works. This answers an open problem mentioned in the work of Hopkins et al.
>
> **3. Correctness of Theorem A.3:** Thank you for your skepticism, but as we will explain the result in Theorem A.3 is correct (though we should note there is a typo on page 11 line 558, where it should say $L(*,\cdot)=1$). In the example you constructed, we suspect there may be a typo in your definitions, since if the distribution $\mathcal{D}$ is supported on $(b,0)$, then according to your loss definition this distribution would be realizable by hypothesis $g$, since $L(g(b),0) = L(1,0) = 0$ (your loss has $L(1,0)=0$). We suspect your intention in the construction might have been to construct a scenario where the realizable learner always returns a function with loss $1$.  Such cases can indeed occur, and are compatible with our result, since then the best-in-class loss $L_{\mathcal{D}}(\mathcal{C}) = 1$, and we trivially satisfy the agnostic learning guarantee $L_{\mathcal{D}}(A(S)) \leq 1 \leq L_{\mathcal{D}}(\mathcal{C}) + \epsilon$.
>
> We hope this rebuttal convinces you of the significance and novelty of our contributions, and addresses your concern about the correctness of Theorem A.3.

---

> > ### Comment · Reviewer_ajtt · 2025-04-01
> >
> > 1. If a big part of your novelty is that your reduction is representation preserving, you should at least define what you mean by representation preserving in your paper. As far as I can tell, you do not, and it was not at all obvious to me what you meant by it.
> >
> > 2. I'm not sure what version of Hopkins et al you are looking at, but I was unable to find the version of the open question you quoted or anything equivalent in it. But again, Hopkins et al is largely concerned with the distribution-family setting for which your results definitely do not resolve anything.
> >
> > 3. I believe my counter-example had a typo (now fixed) where I swapped the 0 and 1 values of the loss function. But it is made irrelevant by your typo on line 558. I am now just left to wonder what the point of the point of the result is in the first place. Is learning with partial classifiers just equivalent to learning where there happens to be an extra output called * which is never right? If so, what is the point of the model? Wouldn't *any* reduction from realizable to agnostic learning automatically also hold for partial classifiers?

---

> > > ### Author Response · Authors · 2025-04-02
> > >
> > > We thank the reviewer for dedicating their time to reassess our work. Below, we address the new comments provided by the reviewer.
> > >
> > > - The definition of the representation-preserving property is implicit in all of our formal definitions. For instance, see Definitions 3.1 and 3.2. In fact, the reason for introducing a hypothesis class $H$ in addition to the concept class $C$ is to capture this property. However, we acknowledge the reviewer’s comment and will ensure that this is discussed more clearly in the camera-ready version. In short, if the output of a realizable PAC learning algorithm lies in a hypothesis class $H$, our reduction guarantees that the output of the agnostic PAC learning algorithm also lies in $H$.
> > >
> > > - In the final version of the work by Hopkins et al., published in TheoretiCS 2024, they mentioned this open problem. This version is available on arXiv: https://arxiv.org/pdf/2111.04746. Here is the relevant paragraph: "Finally, we note there are a few settings where Algorithm 1 runs into issues, especially discrete infinite settings such as infinite multi-class classification and properties such as privacy that require more careful data handling. We leave the extension of our method to these settings as an intriguing open problem." We note that, prior to our work, there was no agnostic-to-realizable reduction for the multiclass setting with infinite label spaces that also preserves representation. Thus, our work is the first to provide a proof that, in the multiclass setting with an infinite label space, a concept class $C$ is agnostically PAC learnable by learners within a hypothesis class $H$ if and only if $C$ is realizably PAC learnable by learners within the hypothesis class $H$.
> > >
> > > - Your understanding of the partial concepts setting is correct. In addition to $\star$ never being a correct prediction, the other difference is that the symbol $\star$ will never appear as a label in the data. This subject was developed in the work of Alon, Hanneke, Holzman, and Moran (FOCS 2021, https://arxiv.org/abs/2107.08444), with the purpose of unifying a variety of learning scenarios whose learnability is not captured by traditional PAC learning theory that previously required separate problem-specific analyses. One classic example is the problem of learning a linear classifier with a margin. For instance, both the Perceptron and SVM operate over the concept class of all linear classifiers, whose VC dimension grows linearly with the Euclidean dimension. But even in infinite-dimensional spaces, if the data are separable with a margin $\gamma > 0$, the sample complexity remains finite and scales linearly in $1 / \gamma^2$ (supposing the data are normalized, for simplicity). This fact is perfectly captured by noting the VC dimension of the partial concept class of $\gamma$-margin linear classifiers is $1 / \gamma^2$ (the partial concepts in this class are specified by linear classifiers, and output $\star$ for points $\gamma$-close to the separator). They show that, for {0,1,$\star$}-valued partial concept classes, the VC dimension still characterizes PAC learnability. Moreover, there exist partial concept classes that are PAC learnable, yet any disambiguation of them—intuitively, filling in the $\star$ symbols with labels to form a total concept class—is not PAC learnable with the traditional PAC learning framework. We refer the reviewer to the aforementioned paper for a more detailed discussion. Regarding the final question, consider the task of agnostically learning the partial concept class of linear classifiers with margin, under a distribution that does not have a margin (which is admitted in the agnostic setting). If we try to apply the basic reduction from Hopkins et al., there is no realizable labeling of the unlabeled data set, so the realizable learner will not be well-behaved if we just run it on all labelings of the unlabeled data. Thus, it becomes necessary to consider sub-samples of the data, as in our work (Hopkins et al. also consider a variant for partial concepts, which considers all realizable labelings of all subsets of the data, but as we previously mentioned, it still runs into problems in the multiclass partial concepts setting). The main contribution of the current manuscript in this context is proving a representation-preserving reduction that applies even in the general case of multiclass partial concepts with an infinite label space, as well as other settings.
> > >
> > > We hope this rebuttal convinces you further of the significance and novelty of our contributions.
> > >
> > > Once again, thank you for dedicating your time to reassess our work.

---

### Official Review · Reviewer_eCfp · 2025-03-09

**Overall Recommendation:** 4

**Summary:**

The paper studies a representation preserving agnostic to realizable reduction. The reduction can nicely be described as, splitting the training data into two parts $V$ and $T$. Now on the first part of the training data $V$ the learner runs on all subsets the realizable learning algorithm, getting $ 2^{|V|} $ my outputs of the realizable learning algorithm. Now using $ T $ as a validation set the learner picks the best of the $ 2^{|V|} $ hypotheses, created in the first step of the process. The proof follows from a high level from first considering $V$ samples from a mixture of a distribution realizable/labelled by the best reference hypothesis $ c^* $ and a part not realizable by $c^*$. Since the first step runs the realizable algorithm on all subsets, it especially runs on the largest subset $ S' $ of $ S $ that is labelled by $ c^*.$ Thus on this subset the realizable learner gets a good preformance on the part of the distribution labelled by $ c^* $  and since the agnostic algorithm has to compete with $ c^* $ and it fails on the other part of the distribution, the output of the realizable learner is close to the preformance of $ c^* $ on the true distribution. Now to ensure that the output of the realizable learner on the distribution realizable by $ c^* $ is small, the paper makes sure that $ |V| $ is large enough such that the largest subset $ S' $ of $ S $ label by $ c^* $ contains sufficently many examples to get small error under the part of the true distribution realizable by $c^*$.     The next step is to extract this good hypothesis from the $ 2^{|V|},$ hypothesis created in the first step of the algorithm. By Chernoff and a union bound over the $ 2^{|V|} $  hypothesis, the true risk is close to the empirical risk for all the $ 2^{|V|} $ hypotheses and the hypothesis with the smallest empirical risk can be chosen as the final classifier. The above algorithm is related to that presented in Hopkins et al. 2022, but can handle infinite labelspaces as with for instance multiclass classification, which was left as an open problem in Hopkins et al. 2022. Furthermore the framework also works for list learning and multilabel PAC learning (again with infinite labels). Furthermore, the paper shows that the sample complexity of the framework is better when considering the multiclass classification problem (with infinite labels) with Tsybakov and Massart Noise.

**Claims And Evidence:**

Most of the proofs of the claims are in the main, and to the best of my knowledge, these are sound. I did not check the proofs in the appendix.

**Essential References Not Discussed:**

I think it would be good to add a remark about the samples complexity obtained in Brukhim et al. 2022 when presenting corollary 4.3 and the sample complexity obtained in Charikar and Pabbaraju, 2023 when presenting corollary 4.4, since they are a $ 1/\varepsilon $ polynomial factor better. Also, is there any work on multilabel classification, that would also be nice to add.

**Experimental Designs Or Analyses:**

No experiments are included in the paper, so I did not know how to evaluate this question.

**Methods And Evaluation Criteria:**

No experiments are included in the paper, so I did not know how to evaluate this question.

**Other Comments Or Suggestions:**

Congratulations on your nice paper, here are some notes that I took while reading.

- page 3 second column line 131-133 Vapnik 2006, the date might be of?
- page 3 second column line 160 and proof of lemma 4.2, the notation  $ (\mathcal{X} \times \mathcal{Y})^* $ is used differently, as respectively all possible sequences and examples where $ c^{*} $ is realizable.
- page 5 first column line 254-255 is it $ \varepsilon/(2\mu^*) $   or is it $ \varepsilon \mu^*/2 $? That was a general question I had for this proof.
- page 7 first column, from equation 8 to equation 9, where did the $ \epsilon/2 $ from the left side of equation 8 go?

**Other Strengths And Weaknesses:**

The paper is well written. The paper solves an interesting open problem from Hopkins et al. 2022, extending the agnostic to realizable framework to the channeling infinite label space setting, in an elegant way.

**Questions For Authors:**

1. I was a bit confused about some of the motiviation but was not able to make it concise, here was my thoughts: Page 2 first column line 055-073: Looking at David 2016 and the proof of their theorem 3.3, my understanding of the agnostic sample compression scheme they use is that it is the realizable sample compression run on the largest sub sample $ S' $ of $S$ which is realizable by a hypothesis in $C$. Thus, the output space of their agnostic sample compression scheme is the same as the output space of the realizable sample compression scheme, so representation preserving in terms of the output of the realizable sample compression scheme. Jumping to page 5 second column line 254-268, the framework is applied with algorithm 1 in Brukhim et al. 2022, which from my understanding is an algorithm/realizable sample compressions scheme(please correct me if i am wrong), thus the output of the algorithm 1 presented in this paper is the output of algorithm 1 in Brukhim et al. 2022, so a realizable sample compressions scheme run on some subset of the data set $ S,$. Now again going to Brukhim et al. 2022 and considering their proof of the agnostic case, it uses theorem 3.3. in David 2016, thus it runs the algorithm 1 in Brukhim et al. 2022/the sample compressions scheme on the largest sub sample $ S' $ realizable by the hypothesis in $C$, thus essentially having the same output of hypothesis?. Thus, going back to page 2, first column line 055-073 I don't quite get the motivation for outputting simple learners? It is also known from, for instance, "Optimal Learners for Multiclass Problems" Amit Daniely, Shai Shalev-Shwartz, theorem 1 that in for instance the multiclass learning setting that simple learners/proper learners can not guarantee to learn. My understanding is that the picture is the same for list learning, and that the proof of Charikar and Pabbaraju, 2023 goes through a similar sample compression technique, this understanding is from their section 7.2 I did not check their appendix B. I get that the reduction presented in the paper, in the case that the learner is simple, proper, or close to proper, the reduction will output something simple. However, from my understanding, the cool thing about this paper is that it captures cases with infinite label spaces that the work of Hopkins et al. 2022 did not, e.g., Multiclass classification as a prime example. Thus i don't get this motivation on page 2 first column line 055-073 about necessarily being related to simple learners. I apologize in advance if I have misunderstood some of the previous works since they are not simple for me to grasp. I also understand that the motivation of a work is somewhat surjectiv so as written above I most likely are misunderstanding something.

2. page 5 second column line 265-268 and page 6 first column line 287-292, in both place i think it would be worth mentioning the agnostic abounds obtained in respectively Brukhim et al. 2022 and Charikar & Pabbaraju, 2023, which from my understanding is a $ 1/\varepsilon $ factor smaller.

I would like to keep my score. I also hope that the authors will add a the comment to corollary 4.3 and 4.4 about the worse sample complexity compared to these works and that they in these two cases output the same as the sample compression scheme of David et al. 2016.

**Relation To Broader Scientific Literature:**

The paper presents related work, where the most related is the work by David et al. 2016 and Hopkins et al. 2022. They solve an open problem stated in Hopkins et al. 2022 about giving an agnostic to realizable reduction of the infinite labels multiclass classification setting.

**Theoretical Claims:**

I checked the proofs in the main and to the best of my knowledge they are sound. I did not check the proofs in the appendix.

---

> ### Author Rebuttal · Authors · 2025-04-01
>
> We thank the reviewer for dedicating their time to assess our work. In particular, we thank the reviewer for taking the time to verify that the work is technically sound. We are delighted that the reviewer found that we solved an interesting open problem in an elegant way, and moreover mentioned that our paper is well written. We will make sure to correct the typos and incorporate minor suggestions mentioned by the reviewer for the camera-ready version. Below, we address the major comments provided by the reviewer.
>
> **1. Other comments or suggestions:**
>
> * It is $\epsilon/(2 \mu^*)$.
> * There should be a $\epsilon/2$ on the right-hand side of equation 9.
>
>
> **2. Question about representation preserving property:** Thank you for this insightful discussion. Your understanding of the prior literature is accurate. For multiclass classification, if we use Algorithm 1 from Brukhim et al. (2022) as our input realizable learner, our approach will yield the same hypothesis class as presented in their paper. And indeed, since that algorithm is compression-based, the reduction of David et al. will be representation-preserving.  However, a key advantage of our reduction algorithm lies in its flexibility. For instance, if a realizable learner with simpler hypotheses is available for a given problem, our Algorithm 1 can directly leverage it to produce a correspondingly simpler agnostic learner, which is not necessarily achievable with prior methods.  As another interesting example, our reduction would be applicable, and representation-preserving, for realizable learners based on optimally orienting the one-inclusion hypergraph (which we generally would not expect to be compression-based), and in fact would return a predictor which itself is expressed as such an orientation using a subset of the data $V$.  The fact that we return a predictor produced by the realizable learner is itself another notion of simplicity (e.g., a predictor based on an orientation of the realizable-case one-inclusion hypergaph is simpler than majority votes of such predictors, and the simplicity is even more apparent compared to a predictor based on an orientation of the agnostic one-inclusion hypergraph).

---

### Official Review · Reviewer_N7MP · 2025-03-14

**Overall Recommendation:** 4

**Summary:**

The authors study agnostic learning with black-box realizable learners, extending the work of Hopkins et al. (2022).
They adapt the simple reduction from Hopkins et al. in a very general PAC learning setting (encompassing list learning and many more). They prove that their reduction algorithm achieves a sample complexity of roughly $\frac{\text{realizableAlgo}(\epsilon,\delta) + \log(1\delta) }{\epsilon^2}$. They prove that the $1/\epsilon^2$ is in general unavoidable.
Their algorithm resolves the open problem of Hopkins et al., of providing such a back box reduction for the multi-class case with infinite labels.
Furthermore, they also study particular noise settings.

**Claims And Evidence:**

Yes.

**Essential References Not Discussed:**

All good.

**Experimental Designs Or Analyses:**

N/A.

**Methods And Evaluation Criteria:**

N/A.

**Other Comments Or Suggestions:**

Please check your usage of \citep vs \citet. E..g, use "work of Hopkins et al. (2022)"  instead of "work of (Hopkins et al, 2022)".

Typos:
* "an unified" --> "a unified"
* " (Agnostic → Realizable )." remove space after realizable
* Inconsistent "class vs Class" in "Concept Class C, Hypothesis class H" in Algorithm 1.

**Other Strengths And Weaknesses:**

The paper is nice, continues an important line of work, and should be interesting for the theoretical ICML community. It is somewhat incremental and the new achievements compared to Hopkins et al. (2022) are not fully clear. Please, see questions below.

Some more discussion with other recent attempts combining learners to achieve optimal learners would be nice. The cited Aden-Ali et al paper and others. While not necessarily related to agnostic learning, the general theme of trying to combine basic learners in a simple way seems is quite common nowadays (in contrast with boosting, sample compression, one-inclusion graph-based, etc. techniques).

**Questions For Authors:**

There seems to be a significant difference to Hopkins et al (2022). Note how the reduction calls the algorithm $A(\cdot)$ with potentially non-realizable subsamples. Standard realizable algorithms cannot necessarily handle these. E.g., the Perceptron algorithm might not even stop when run on a non-realizable sample. In some sense you are assuming that the realizable algorithm can detect that the sample is not realizable and then just returns some default hypothesis. Please make this distinction clearer in the next version. Or is there an easy fix? E.g., in Hopkins et al (2022) an unlabeled sample is used an labeled only with hypotheses, hence realizability is guaranteed.

It is not really clear from the paper if the new results (e.g., reduction for the $|Y|=\infty$ case) require the adapted blackbox reduction or if an new analysis of the original algorithm of Hopkins et al. (2022) would suffice. Please clarify. Are the changes necessary?

Also, Theorem 2.2 states that their reduction needs overall $1/\epsilon^2$ samples for VC classes (instead of the worst-case $1/\epsilon^3$). Is this also possible here? Corollary 4.3 here says that the new reduction requires $1/\epsilon^3$ overall at least for the DS dimension case. What about finite or binary $Y$?

The tightness proof is not very satisfactory. It only holds for the $O(1)$ vs $1/\epsilon^2$ case. In particular it is not shown that an $1/\epsilon^3$ can ever be required for this reduction (in some natural case like multi-class, preferably with finite labels).

**Relation To Broader Scientific Literature:**

Some clearer distinction with Hopkins et al. (2022) would be nice. See below.

**Theoretical Claims:**

Proofs are mostly standard and correct.

---

> ### Author Rebuttal · Authors · 2025-04-01
>
> We thank the reviewer for dedicating their time to assess our work. In particular, we thank the reviewer for taking the time to verify that the work is technically correct. We are delighted that the reviewer found our work nice and important, and moreover mentioned that our paper should be interesting to the theoretical ICML community. We will make sure to correct the typos and incorporate minor suggestions mentioned by the reviewer for the camera-ready version. Below, we address the major comments provided by the reviewer.
>
> **1. Running realizable learner on unrealizable sets:** Thank you for pointing this out. In this work, our definition of learning algorithm requires it to always return a function.
>  However, for learners that only satisfy this for realizable input data sets, we can still apply our reduction by simply modifying line 2 of our algorithm to only run $A$ on all subsets of $V$ that are realizable by $\mathcal{C}$ (which could be identified using a "weak consistency" oracle for $\mathcal{C}$).  The theorem and proof remain valid.
>
> **2. The algorithm of Hopkins et al. does not work for multiclass classification with an infinite effective label space:** Thank you for this insightful question. We address it by noting the Hopkins et al. reduction fails for the well-studied "stars and sets" concept class, namely Example 4.1 of Daniely et al. (2011). Specifically, consider $\mathcal{X} = [0,1]$, denote by $\mathcal{F}(\mathcal{X})$ the collection of all finite subsets $A\subseteq \mathcal{X}$. Let the label space $\mathcal{Y} = \mathcal{F}(\mathcal{X})  \bigcup  \lbrace\star\rbrace$, where $\star$ is a special label (not to be confused with "$*$" from the partial concept classes section). For every $A\subseteq \mathcal{X}$, define $f_{A}:\mathcal{X}\rightarrow\mathcal{Y}$ as follows:
> $$
> f_{A}(x) = \begin{cases}
> 		      \star & \text{if } x\in A \\\\
>               A & \text{otherwise}.
> 		   \end{cases}
> $$
> Let the concept and hypothesis class be $\mathcal{C} = \mathcal{H} = \lbrace f_A: A\in \mathcal{F}(\mathcal{X}) \cup \mathcal{X}\rbrace$. Then, there is a realizable learner $\mathcal{A} _ {\text{good}}$ that returns $f_{\mathcal{X}}$ unless a label $A \in \mathcal{F}(\mathcal{X})$ appeared in the sample, in which case it returns $f_A$. Let $\mathcal{D}$ have marginal on $\mathcal{X}$ the uniform distribution over $\mathcal{X}$, and let the labels be always $\star$ (i.e., realizable with target $f_{\mathcal{X}}$). Now, consider the algorithm of Hopkins et al. with this scenario and realizable learner $\mathcal{A} _ {\text{good}}$. Let the unlabeled dataset be $S_U = \lbrace x_1, x_2,\cdots, x_n\rbrace$, and the labeled dataset be $S_L =\lbrace(x_{n+1},\star),(x_{n+2},\star),\cdots,(x_{n+m},\star)\rbrace$, and with probability one these $x$'s are all distinct. Denote $A = \lbrace x_{n+1},x_{n+2},\cdots,x_{n+m}\rbrace \in \mathcal{F}(\mathcal{X})$. Then their algorithm would run $\mathcal{A} _ {\text{good}}$ on all realizable labelings of $S_U$; in particular, one of these is $(S_U,f_A(S_U)) = \lbrace (x_{1},A),\ldots,(x_{n},A)\rbrace$, and the output hypothesis of $\mathcal{A} _ {\text{good}}(S_U,f_A(S_U))$ would be $f_A$. By the definition of $f_A$, we know that the empirical error of $f_A$ on $S_L$ is 0.  Their algorithm then outputs any ERM on $S_L$ from these functions produced by $\mathcal{A} _ {\text{good}}$, which means their algorithm can output $f_A$. However, the true error rate of $f_A$ is 1, while the best error in the concept class is 0. Thus, their algorithm fails for this concept class (for essentially the same reason ERM fails for this concept class).  In contrast, our algorithm returns $f_{\mathcal{X}}$, hence achieves error 0.
>
> **3. Sample complexity when we have a finite effective label space:** Thank you for pointing this out. For multiclass classification with a finite effective label space size $|\mathcal{Y}| _ {\text{eff}}$, the sample complexity of our reduction can be reduced to $\tilde{O}(1/\epsilon^2)$ rather than $\tilde{O}(1/\epsilon^3)$. More specifically, in this case learnability of $\mathcal{C}$ is equivalent to finite Natarajan dimension $d$, and we can use a similar argument as above, only running the realizable learner on the maximal realizable subsets of $V$, of which there are at most $|V|^d |\mathcal{Y}|^{2d}_{\text{eff}}$ using Natarajan's generalization of the SSP Lemma, so that this bounds the size of $\mathcal{H}_V$ (rather than $2^{|V|}$), leading to a refinement of the final bound (though this analysis then becomes much closer to that of Hopkins et al.).
>
> **4. Sample complexity lower bound:** We note that in the case of a finite effective label space, we are able to prove an improved sample complexity of $\tilde{O}(1/\epsilon^2)$. On the other hand, we agree with the reviewer that exploring the possibility of sharpness of the $\tilde{O}(1/\epsilon^3)$ bound in future work would be interesting.

---

### Official Review · Reviewer_rut2 · 2025-03-22

**Overall Recommendation:** 4

**Summary:**

This paper studies the PAC learning of the problem of multiclass classification with unbounded numbers of labels. The primary contribution is a novel reduction from the agnostic learning setting to the realizable setting that preserves the structure of the output space, which resolves an open problem posed by Hopkins et al., 2022. The authors introduced "unified PAC learning" which encompasses multiclass, list, ad multilabel PAC learning. Additionally, they explores reductions under Massart and Tsybakov noise and extend their results to partial concept class.

**Claims And Evidence:**

Yes.

**Essential References Not Discussed:**

No.

**Experimental Designs Or Analyses:**

Yes.

**Methods And Evaluation Criteria:**

Yes.

**Other Comments Or Suggestions:**

No.

**Other Strengths And Weaknesses:**

The paper is well written. The mathematical development is clear and concise.

**Questions For Authors:**

No.

**Relation To Broader Scientific Literature:**

It's very related.

**Theoretical Claims:**

No.

---

> ### Author Rebuttal · Authors · 2025-04-01
>
> We thank the reviewer for dedicating their time to assess our work. We are delighted that the reviewer found our algorithm novel, and mentioned that our paper is well written.
>
> We would be happy to provide additional clarification on any aspect of the paper that could help inform the review score.

---

### Decision · Program_Chairs · 2025-05-01

**Decision:**

Accept (poster)

**Comment:**

This paper presents a simple reduction scheme from agnostic to realizable learning for multiclass classification with potentially infinite label space. The approach is also applied to list PAC model and multi-label learning. While the work shares high level insights with Hopkins et al, this paper presents new techniques that lead to much simpler analysis and more general applications of the meta theorem (for example, the main results hold for any distribution and while Hopkins et al needed certain distributional assumptions).